METHODS AND RESOURCES

# Pathogenic missense protein variants affect different functional pathways and proteomic features than healthy population variants

**Anna Laddach**[☯‡¤]**, Joseph Chi Fung Ng**[☯‡]**, Franca Fraternali**[*]

Randall Centre for Cell and Molecular Biophysics, King's College London, London, United Kingdom

☯ These authors contributed equally to this work.
¤ Current address: The Francis Crick Institute, London, United Kingdom
‡ These authors share first authorship on this work.
* franca.fraternali@kcl.ac.uk

**Data Availability Statement:** All relevant data of tables and figures of this paper is available in the Supporting Information files. The ZoomVar database, accessible at fraternalilab.kcl.ac.uk/

## Abstract

Missense variants are present amongst the healthy population, but some of them are causative of human diseases. A classification of variants associated with "healthy" or "diseased" states is therefore not always straightforward. A deeper understanding of the nature of missense variants in health and disease, the cellular processes they may affect, and the general molecular principles which underlie these differences is essential to offer mechanistic explanations of the true impact of pathogenic variants. Here, we have formalised a statistical framework which enables robust probabilistic quantification of variant enrichment across full-length proteins, their domains, and 3D structure-defined regions. Using this framework, we validate and extend previously reported trends of variant enrichment in different protein structural regions (surface/core/interface). By examining the association of variant enrichment with available functional pathways and transcriptomic and proteomic (protein half-life, thermal stability, abundance) data, we have mined a rich set of molecular features which distinguish between pathogenic and population variants: Pathogenic variants mainly affect proteins involved in cell proliferation and nucleotide processing and are enriched in more abundant proteins. Additionally, rare population variants display features closer to common than pathogenic variants. We validate the association between these molecular features and variant pathogenicity by comparing against existing in silico variant impact annotations. This study provides molecular details into how different proteins exhibit resilience and/or sensitivity towards missense variants and provides the rationale to prioritise variant-enriched proteins and protein domains for therapeutic targeting and development. The ZoomVar database, which we created for this study, is available at fraternalilab.kcl.ac.uk/ZoomVar. It allows users to programmatically annotate missense variants with protein structural information and to calculate variant enrichment in different protein structural regions.

ZoomVar, hosts the underlying variant enrichment statistics presented here and provides functionalities freely available for public query.

**Funding:** This research was supported by the British Heart Foundation (https://www.bhf.org.uk/, RE/13/2/30182 to FF and AL), Croucher Foundation Hong Kong (https://croucher.org.hk/, scholarship to JCN), the Medical Research Council (https://mrc.ukri.org/, MR/L01257X/1 to FF) and the Biotechnology and Biological Sciences Research Council (https://bbsrc.ukri.org/, BB/T002212/1 to FF and JCN). The funders had no role in study design, data collection and analysis, decision to publish, or preparation of the manuscript.

**Competing interests:** The authors have declared that no competing interests exist.

**Abbreviations:** EGFR, epidermal growth factor receptor; GPCR, G protein-coupled receptor; GSEA, Gene Set Enrichment Analysis; MAF, minor allele frequency; NPIP, Nuclear pore complex interacting protein; NUT, Nuclear Testis protein; PCA, Principal Component Analysis; PTM, posttranslational modification; TSG, tumour suppressor gene; VES, Variant Enrichment Score.

## Introduction

The genomic revolution has brought about large advances in the identification of disease-associated variants. However, despite the recent explosion of genetic data, the problem of "missing heritability" still persists [1], where the genetic component of a phenotype remains poorly identified. For some variants, a causal link to the disease in question is difficult to establish, e.g., variants with low penetrance and/or those with higher penetrance which are unique to single/few individuals, such as de novo variants implicated in developmental disorders [2]. Compensated pathogenic mutations represent another such case, where the pathogenic effects of a mutation are negated by another variant [3]. Difficult cases also arise in the analysis of somatic cancer variants, where driver mutations can be challenging to segregate from passenger mutations; moreover, this classification may vary from case to case [4]. These variants pose challenges to the detection of disease association using existing statistical methods. In head-to-head comparisons against large-scale saturation mutagenesis screens, where mutational impact could be measured in vitro, current predictive methods were shown to be limited in accuracy [5,6]. Moreover, the majority of in silico methods focused primarily on detecting differences between disease-associated and common variants, thus it has been suggested that these methods do not perform so well when distinguishing rare neutral variants from those which are pathogenic [7]. The boundary separating disease-causing from neutral variants can be fluid: for example, a number of missense variants thought to lead to severe mendelian childhood disease were identified in nominally healthy individuals in the ExAC database [8].

In silico variant impact prediction methods often fall short at explaining the mechanisms which lead to the true biological impact of a given missense variant, which depends on the protein—and position within the protein—to which a missense variant localises. With measurements of the abundance of transcripts and proteins across tissues and cellular states becoming more widely available [9,10], attempts to map consequences of variants on gene/protein expression have quickly emerged [9,11]; however, the use of transcript/protein abundance in understanding the distribution of variants is underexplored. The abundance and stability of transcripts and proteins conceivably constrains the amount of variations tolerable to the given gene/protein [12,13]; however, this has not been explored systematically using new, large datasets (e.g., [10,14,15]).

A large-scale analysis which surveys the distribution of variants across proteins with different proteomics profiles, as well as regions within proteins defined using annotations such as protein disorder and solvent accessibility, is essential to derive molecular principles and rules which dictate the pathogenicity of a given variant. However, the exploration of these molecular features in the context of variant annotation has often been limited to individual pathways and specific sets of proteins. For example, certain protein subsets have been used as controls to validate variant impact prediction tools, e.g., olfactory receptors are a control case absent of disease-causing variants [7,16]; the enrichment and depletion of variants in cancer driver genes, specifically within protein interfaces [17], as well as protein regions defined by solvent accessibility [18,19], have also been extensively investigated. Only a small number of studies have attempted to compare the distributions of selected datasets of pathogenic and nonpathogenic variants [20,21,22], but these focused on specific protein features, and none used a unified approach to consider multiway comparisons between pathogenic variants and those with different frequencies within the population. While smaller-scaled studies have provided tools and methods to explore variant distributions across proteins, to our knowledge, a systematic approach which achieves the following is yet to exist: (1) compares comprehensive sets of pathogenic and nonpathogenic variants; (2) considers a wide range of features such as protein structure, pathways and measurements of transcripts, and protein abundances; and (3) utilises

robust statistical methods which take into account the substantial differences in protein size and data coverage (e.g., systematic biases in availability of crystal structures). Methods which address these concerns are in great need to decipher molecular principles which underlie variant pathogenicity.

Here, we present a detailed analysis of different classes of missense variants, including germline disease variants, somatic cancer variants (both "driver" and "passenger" variants with varying effects on tumour progression), as well as population variants of different frequencies, in the quest to extract the governing principles of variant pathogenicity. We rely on the synergy between utilising 2 types of data: First, we place emphasis on mapping the localisation of variants on protein structures, taking into account their positions in the protein fold, as well as their proximity to functional sites (e.g., posttranslational modifications (PTMs) [20,21,23–26]. Such protein structural information has been shown to be effective in uncovering the impact of variants at the molecular level [27]. In the field of cancer research, protein structure-based methods have been used to successfully predict cancer driver genes, as validated by a recent large-scale study by Bailey and colleagues [28]. Second, we also make use of recently available large-scale proteomic measurements, including protein abundance [10], half-life [15], thermal stability [14], and transcriptomics data [9], to uncover biophysical and biochemical principles governing the impact of variants. Our analyses highlight a striking difference in the enrichment of pathogenic and population variants, which depends upon their localisation to protein domains and structural features. This integrative analysis provides molecular details into how resilience and sensitivity to missense variants are manifested in different proteins and functional pathways. We have created the ZoomVar database (fraternalilab.kcl.ac.uk/ZoomVar), which holds the data generated in this analysis. ZoomVar is designed for large-scale programmatic structural annotation of missense variants and calculation of the enrichment of missense variants in different protein structural regions. Comprehensive mapping of structural localisation of variants could inform the development of therapeutic interventions, e.g., structure-based drug design and/or drug repurposing [29]. More generally, the wealth of features that separates missense variants in health and disease could contribute to charting the biophysical rules which govern missense variant impact.

## Results

### A detailed protein-centric anatomy of variant enrichment across scales

We present a multifactorial analysis of missense variants observed in the general population (gnomAD database) [30], in comparison to somatic cancer-associated missense variants from the COSMIC database [31] and disease-associated missense variants from the ClinVar database [32]. Throughout this analysis, we further divide the gnomAD data by their minor allele frequencies (MAF) into common and rare variants, to investigate whether there are differences between these 2 subsets. We also considered all variants in a continuum of pathogenicity by ranking them using in silico variant impact predictions and evaluate features which are associated with such pathogenicity measures. A summary of the numbers of missense variants investigated is given in Table 1, and a more detailed breakdown is given in S1 Data.

### Defining the protein anatomy

We compare pathogenic and population variants in terms of their associations with specific features across the molecular scale, in a framework we call "protein anatomy," where we partition the human proteome in different ways. This includes the consideration of individual "proteins" (for example, investigating the enrichment of missense variants in the epidermal growth factor receptor [EGFR] protein), specific constituent "domains" of proteins

**Table 1. Numbers of missense variants which localise to different levels and regions of protein anatomy.** Data are listed for each of gnomAD, COSMIC, and ClinVar datasets. Here, "common" and "rare" variants are subset of gnomAD defined using the minor allele frequency (MAF) cutoff of 0.01, below which variants are classified as "rare." This definition is used throughout this work except for the analysis on varying the "rarity" of variants (see main text). Note that the full-length domain-type statistics are omitted here, as by definition they will be identical to the "full-length domain" row. Fig 1B illustrates the definition of regions listed here in the first column.

| | Common | Rare | COSMIC | ClinVar |
|---|---|---|---|---|
| full-length protein | 54,571 | 3,806,698 | 1,731,030 | 21,272 |
| full-length domain | 23,634 | 1,772,768 | 852,597 | 15,831 |
| surf | 12,151 | 966,409 | 491,179 | 8,558 |
| interact | 403 | 38,108 | 22,205 | 768 |
| core | 2,789 | 296,291 | 152,356 | 5,194 |
| intra-ord | 20,650 | 1,575,286 | 755,683 | 14,620 |
| intra-dis | 2,984 | 197,482 | 96,914 | 1,211 |
| inter-dis | 17,352 | 1,045,997 | 439,437 | 1,128 |
| phos | 1,661 | 158,192 | 82,364 | 2,362 |
| ubiq | 440 | 52,250 | 25,778 | 607 |

(e.g., the EGFR tyrosine kinase domain), or generally for all instances of a "domain-type" (e.g., all tyrosine kinase domains) found in the human proteome. These are referred to as the "levels" of protein anatomy (Fig 1A). For each level, we analyse variant enrichment in full-length entities (i.e., protein/domain/domain-type, dependent on the level of interest) and also various constituent "regions" defined using different criteria (Fig 1B and 1C). These include protein structural information (partitioning into surface, core, and interacting interfaces), protein disorder predictions (segregating into ordered and disordered regions, within or outside of Pfam domains), and vicinity to functional sites such as phosphorylation and ubiquitination sites. We explore the interplay of "microscopic," atomistic protein structural features, and large-scale, "macroscopic" features like functional pathways, as well as the various proteomics features collated (see Materials and methods and below), in terms of understanding the distributions of pathogenic and nonpathogenic missense variants over these features.

## Evaluating variant enrichment

To quantify missense variant enrichment, we employ a similar approach to that used in the prediction of cancer driver genes [17]: Variant enrichment has been modelled as the assignment of an observed number of variants into different levels and regions of the protein anatomy. Based on the size (in terms of the number of amino acids) of the entity of interest, we calculated the probability of observing the given number of variants localised to this entity. This quantified the likelihood of observing the given amount of variants over the null scenario where no bias towards localising to any entities were expected, while also taking the size of the region/protein/domain into account. Using the binomial distribution (Materials and methods, Eq 1), this probabilistic calculation yielded a Variant Enrichment Score (VES), ranging from 0 to 1 (Figs 1D and S1 and S2; also see Materials and methods), allowing for intuitive interpretation. For details, please read S1 Text. We also quantified the robustness of such quantification of variant enrichment: The significance of the enrichment/depletion of missense variants, in terms of their density, is assessed by comparison to simulated null distributions, in which the number of missense variants is kept identical to that observed in the data, but their positions within the protein are randomised. This goes beyond similar studies (e.g., [20,21,26]) and addresses biases which could result from the selective focus in molecular studies of disease-related proteins.

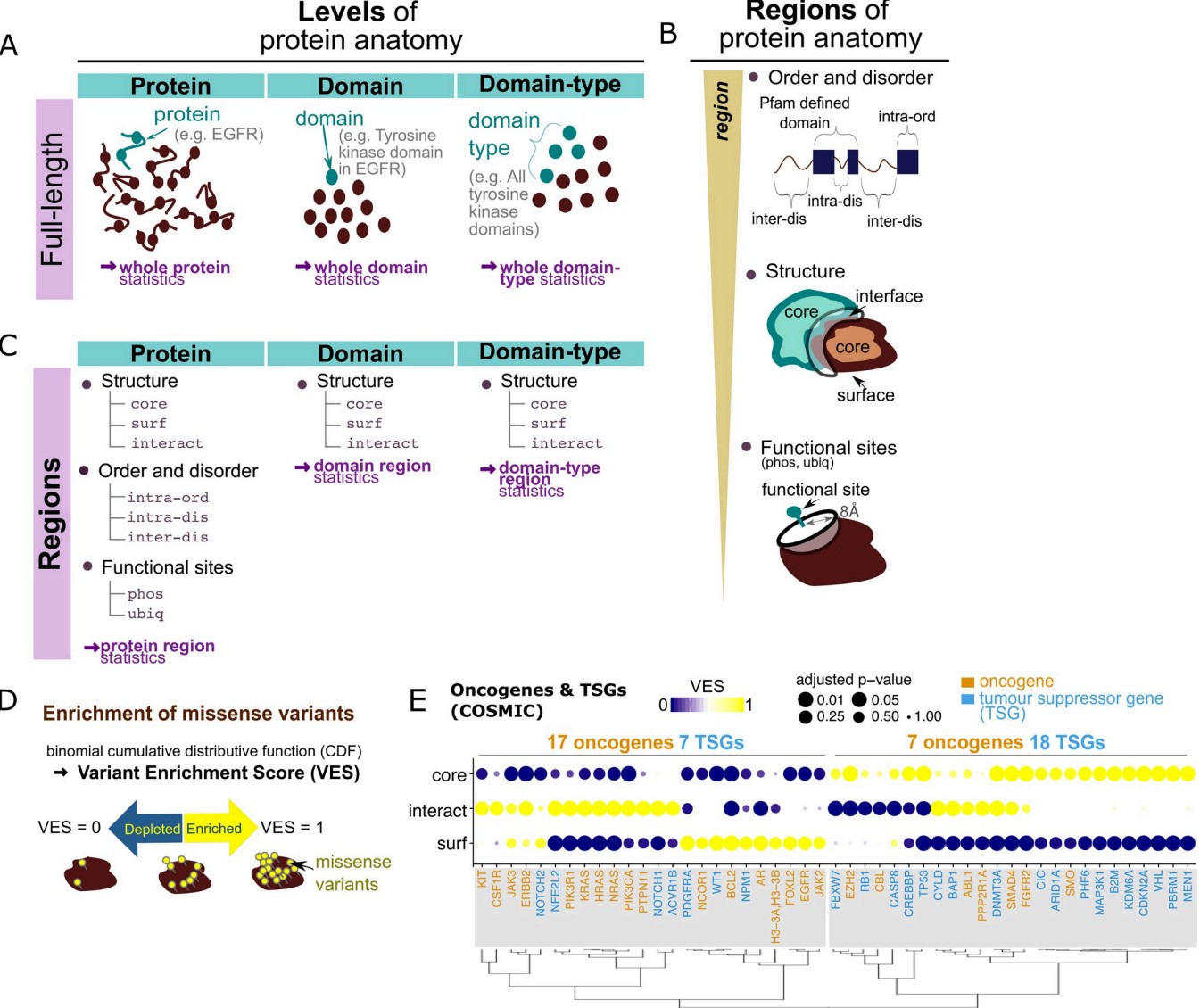

**Fig 1. Variant enrichment crossing scales of protein anatomy.** (A) Levels of the protein anatomy. At the protein/domain level, the number of missense variants in a protein or domain is compared to the number of missense variants in the whole dataset which localise to defined proteins/domains. At the domain-type level, the number of missense variants in a particular Pfam defined domain-type is compared to the total number of missense variants which localise to any Pfam domain-type. These calculations are referred to as the "full-length" protein/domain/domain-type variant enrichment in this manuscript, in contrast with the calculations at regions of protein anatomy defined next. (B) Regions of the protein anatomy. We considered different levels of definition of protein regions, including (i) regions close to functional (phosphorylation/ubiquitination) sites; (ii) structural regions (core, surface [surf], and interface [interact]) of a protein; and (iii) regions predicted to be ordered or disordered which lie either within or outside of Pfam-defined domains. (C) Lists of regions considered at each level of the protein anatomy in this study. (D) The calculation of enrichment at the different levels is statistically assessed using the binomial distribution. The binomial cumulative distributive function constitutes a VES with value range 0 to 1, which quantifies enrichment. (E) Enrichment of COSMIC missense variants in protein core, surface, and interface regions, across a list of annotated oncogene (orange annotations next to the dendrogram) and TSG (blue) products. Size of points denote the level of statistical significance for the calculated VES. The genes were grouped into 2 clusters using hierarchical clustering over the VES statistics (clusters highlighted by grey rectangles; also see dendrogram); the number of oncogenes and TSGs in each cluster is noted. VES statistics can be readily browsed on the online ZoomVar web application. CDF, cumulative distributive function; EGFR, epidermal growth factor receptor; TSG, tumour suppressor gene; VES, Variant Enrichment Score.

## Validating VES calculation

We first applied the method of VES calculation and examined known cases of variant enrichment. By examining a curated list of oncogenes and tumour suppressor genes (TSGs) [33] in

the COSMIC dataset, we found, in agreement with others [25,26,34], that these proteins could be classified into 2 groups by considering their patterns of variant distribution, one comprising proteins enriched in mutations mainly at interaction interfaces and surfaces, and another group in the core (Fig 1E). Some proteins in the latter group also show enrichment in interacting interfaces, but a clear depletion of mutations at the surface is evident. The segregation of these 2 groups in terms of cancer driver status has strong statistical support (Fisher exact test *p*-value = 0.0042): The first group of proteins is mainly (17 out of 24) composed of oncogenes, and the other mainly of TSGs (17 out of 25). These results are consistent with the hypotheses that activating mutations in oncogenes are likely to affect particular functions by perturbing specific interactions, while inactivating mutations in TSGs abrogate protein function [26,34]. This validates the ability of our VES formula to recapitulate known variant-enriched and depleted cases.

## Robust statistical quantification confirms and extends variant enrichment patterns in large variant datasets

We reasoned that with the statistical framework we established, as well as the large variant datasets at hand, we could give robust quantification of the enrichment of variants in health and disease. While some trends of variant enrichment have been previously discussed [17–22], our method enables a unique probabilistic evaluation of these patterns in a uniform manner over ClinVar, COSMIC, and gnomAD datasets.

## Population and disease-associated variants localise to different protein regions

We first analysed the trends in the enrichment of variants in different regions of the protein anatomy, defined using structural information, order and disorder, and the vicinity (distance $\leq 8$ Å) of the variant positions to PTMs (see above).

The following findings are highlighted:

**Different structural localisation of pathogenic versus population missense variants.** We find ClinVar variants to be enriched in both protein cores and interfaces but depleted on protein surfaces (Figs 2A–2C and S3). This reflects the potential disruption, caused by such mutations, of structurally and functionally important sites [20,21,25]. The enrichment of ClinVar variants is further demonstrated by their tendency to affect residues which are highly connected when considering network representations of protein structures (see S3 Text). GnomAD variants (both common and rare) and somatic variants falling outside of cancer-related genes display the opposite trend, as variants tend to localise preferentially to protein surfaces, and are therefore less likely to impact on protein structure and function than either core or interface mutations. Somatic variants in cancer genes follow trends closer to ClinVar variants, with slight, but significant, depletion on the surface, but enrichment in the core. Protein interfaces are enriched in disease-associated variants but depleted of gnomAD rare variants. GnomAD common variants appear neither significantly enriched nor depleted; however, this may result from the relative sparsity and high dispersion of the data (fewer variants are shared between many individuals; see Table 1). Interestingly, COSMIC noncancer gene variants appear depleted in interacting interfaces. However, it becomes clear that they are actually significantly enriched when compared to simulated null distributions (see S3 Fig) and that this enrichment is due to a small subset of proteins which harbour a large number of variants at interface regions. Genes in which these variants reside may be putative driver genes (see S4 Data), as a number of known driver genes are enriched in variants in protein interface regions

## Variant enrichment in protein regions

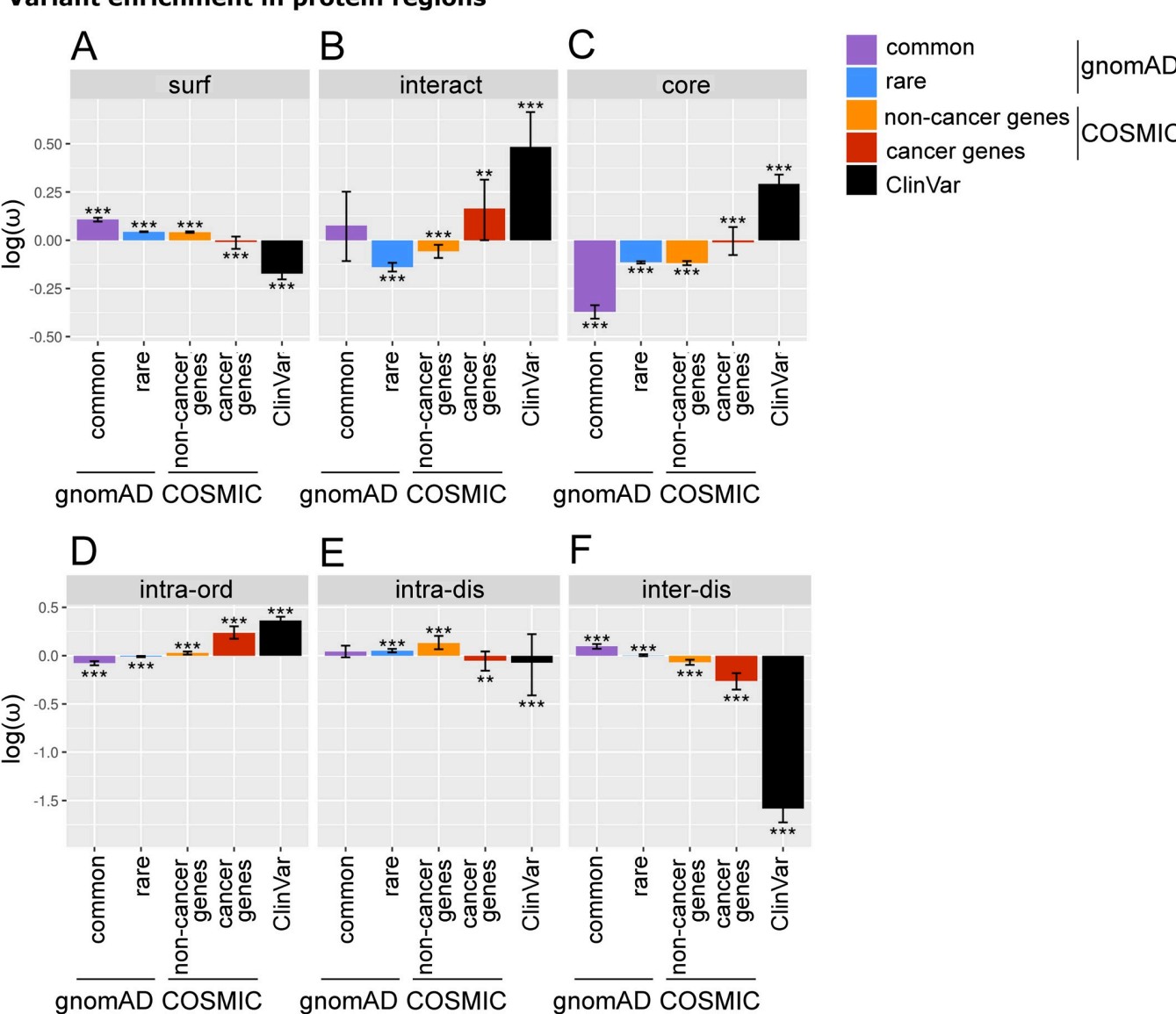

**Fig 2. The localisation of variants to protein regions.** (A–C) The density of mutations in different protein regions, calculated using Eq 1. Density values ($\omega$) were log-transformed such that negative values indicate a depletion of missense variants, while positive values indicate enrichment. Error bars depict 95% confidence intervals obtained from bootstrapping. Significance was calculated by comparison to simulated missense variant distributions (significance level indicated by: * $q$-value < 0.05, ** $q$-value < 0.001, *** $q$-value < 0.0001). Note here the COSMIC set is split into cancer genes (i.e., mutations mapping to proteins found in the COSMIC CGC) and noncancer gene subsets. Data are shown for protein surface (surf, panel A), interacting interface (interact, B), and core (C). (D–F) Density of mutations analogous to panels (A–C) but in regions defined by order and disorder. Data are shown here for intra-domain ordered (intra-ord, panel D), intra-domain disordered (intra-dis, E), and inter-domain disordered (inter-dis, F) regions. See S2 Data for the underlying data. CGC, Cancer Gene Census.

[17,26,28], and this phenomenon has been exploited by Porta-Pardo and colleagues [17] to identify cancer driver genes.

**Pathogenic variants tend to localise to ordered regions within domains.** For variant enrichment in ordered and disordered regions, we again observe clear segregation between disease and population variants (Fig 2D–2F). ClinVar and COSMIC variants are depleted in inter-domain disordered regions and enriched in intra-domain ordered regions. In contrast,

gnomAD variants (both rare and common) appear enriched in inter-domain disordered regions and depleted in intra-domain ordered regions. These results suggest, as one would intuitively expect, that variants are more likely to be pathogenic if they fall within ordered domain regions.

**Pathogenic variants are close to phosphorylation sites.**   When proximity to PTMs is considered (S3 Fig), ClinVar variants appear enriched in terms of the density of missense variants close to phosphorylation sites but not significantly so in comparison to the simulated null background; this may be again due to data sparsity, as suggested by large bootstrapped confidence intervals (S3 Fig). COSMIC cancer gene variants are also close to phosphorylation sites; however, COSMIC noncancer gene variants, which appear depleted in terms of variant density, are also significantly enriched close to phosphorylation sites in comparison to simulated null distributions (S3 Fig). This indicates that, in agreement with a number of other studies [35,36], the disruption of phosphorylation sites may play a particularly important role in cancer. In contrast to phosphorylation sites, all datasets appear depleted of variants close to ubiquitination sites (S3 Fig).

These analyses conclude that the enrichment of missense variants at various structural features consistently segregate population variants from disease-associated ones. For the majority of structural regions defined here, the greatest, most consistent distinction is always seen between common and ClinVar variants, provided that the data are not too sparse. We also observed different patterns of pathway enrichment for variants in surfaces, cores, and interfaces (see S3 Text).

## Towards a domain-centric landscape of variant enrichment

We now proceed from the protein level to examine variant enrichment across domain-types. Agglomerating missense variants at the domain-type level has the advantage of enhancing the statistical power to detect variant enrichment in terms of different protein structural features ([37] and references therein). Here, in contrast to previous studies which focus on variants clustered in sequence or structure space [22,37,38], we survey the landscape of variant enrichment across domain-types and compare the patterns of enrichment of variants from the different health and disease datasets which we have examined above. We focus our discussion on the most variant-enriched domain-types from each of the 4 variant sets. A comprehensive list comprising the union of the top 20 enriched domain-types for each dataset can be found in S4 Fig. Fig 3 shows some selected examples of this list; the missense variant enrichment at the full-length domain-type level (Fig 3A) and in each structural regions (Fig 3B) are depicted. Here, domain-types which are enriched in variants only in the COSMIC and ClinVar datasets can be seen, including known drug targets such as tyrosine kinase (Pkinase_Tyr) and ion channel (Ion_trans) domain-types. A handful of domain-types, which are only enriched in COSMIC variants, include Cadherin_tail and Laminin_G_2 (Fig 3A), both of which play an important role in cancer [39,40].

Some domain-types (e.g., Serpin, UDPGT, Collagen, and EGF_CA) contain variants from all 4 datasets. In such domains, it is likely that the precise structural localisation of a variant determines whether it plays a pathogenic role. Intriguingly, a few domain-types, such as NPIP (Nuclear pore complex interacting protein) and NUT (Nuclear Testis protein) appear only enriched in common variants (Fig 3A). This could suggest that these domains take part in functions for which it is desirable to maintain diversity within a population; however, little is known about either domain-type (Pfam accessions PF06409; PF12881). Therefore, this further highlights the bias in the number of studies targeting domains associated with disease, rather than those enriched in population variants.

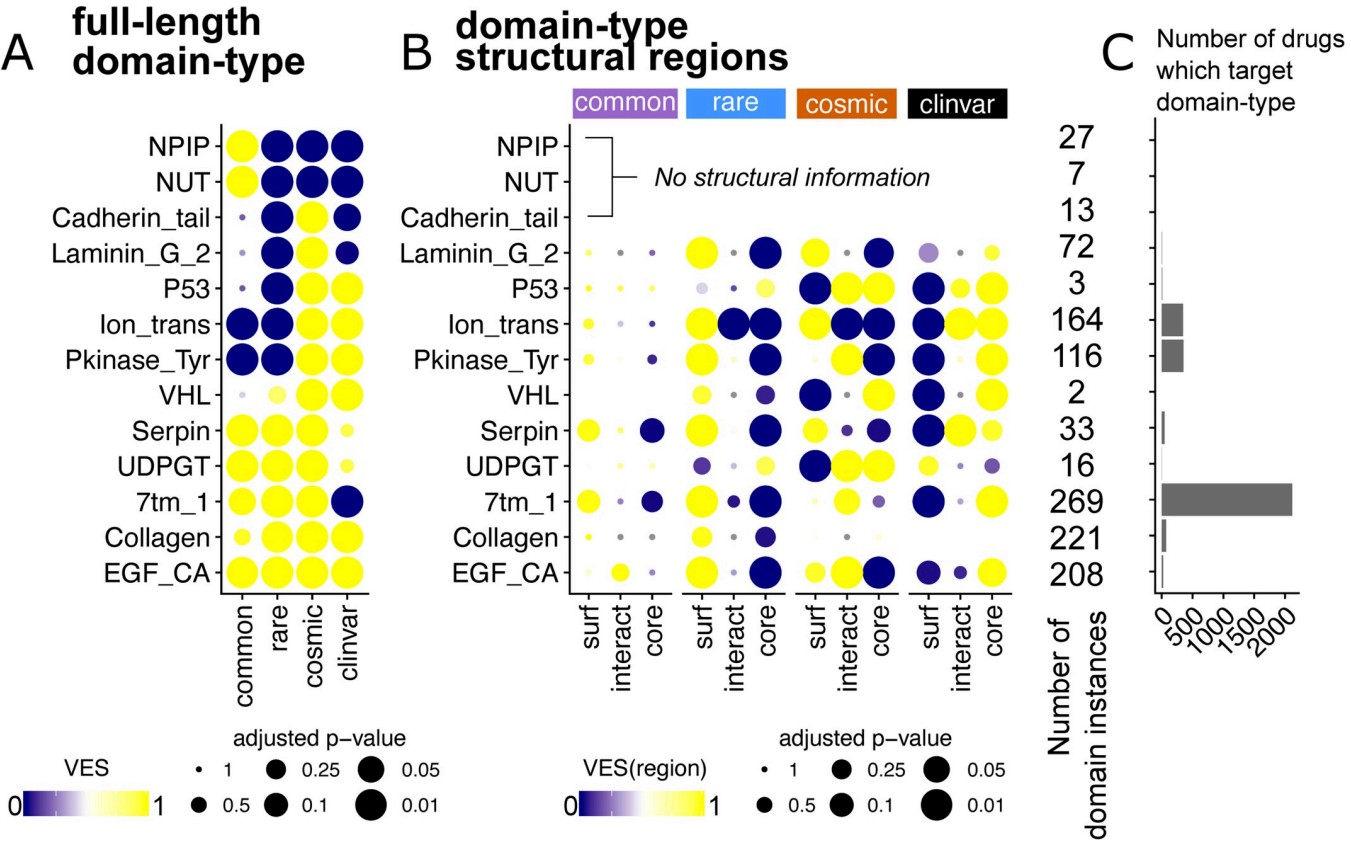

**Fig 3. A domain-centric landscape of variant enrichment.** Here, selected domain-types discussed in the main text are depicted. (A) VES at the full-length domain-type level. Bubble sizes are scaled by the adjusted *p*-value of the VES statistic. (B) VES calculated for each structural region (surface [`surf`], interacting interface [`interact`], and core) for the selected domain-types. (C) The number of drugs known to target proteins containing each domain-type is depicted as a bar graph. See S2 Data for the underlying data. NPIP, Nuclear pore complex interacting protein; NUT, Nuclear Testis protein; VES, Variant Enrichment Score.

It also becomes apparent that the global trends in variant localisation to the core, surface, and interface regions observed above are recapitulated here (Fig 3B) for those domain-types with structural coverage. The majority of domains are enriched in gnomAD (rare and common) variants at the surface but ClinVar variants at the core. For COSMIC, the patterns of localisation are more mixed, but it is clear that in comparison to the gnomAD sets, a larger proportion of domain-types are enriched in COSMIC variants at the core or interface. These include domain-types with known cancer driver associations, such as the P53 and VHL domains [41]. Case studies on CATH architectures [42] and DNA-binding domains further highlight our observed patterns of variant enrichment (see S3 Text).

We also explored how the targeting of domains by drugs and small molecules mapped to the observed landscape of variant enrichment. Using DrugBank [43] data, we observe that the targeting of domain-types by existing drugs is highly biased towards a small number of domain-types (Fig 3C), such as G protein-coupled receptors (GPCRs) and tyrosine kinase, as already extensively pointed out [44]. Indeed, we observe a large number of drugs targeting proteins containing 7tm (GPCR) domains. These domains are enriched in variants from the gnomAD and COSMIC database but are devoid of disease-associated ClinVar variants (Fig 3A). By analysing drug availability together with variant enrichment, this approach allows for more informed decisions in selecting new therapeutic targets. For example, there are domain-types

which could be targeted by few or no drugs but are enriched in COSMIC and/or ClinVar variants. This could offer a starting point to prioritise drug discovery efforts for these domain-types. For domain-types already targetable by drugs, our analysis highlights domains to which multiple disease-associated variants localise, which could give scope for drug repurposing or redesign (see Discussion).

Put together, our statistical method gives probabilistic assessment of variant enrichment and yields robust quantification of these patterns across different protein and domain regions. This enables extending previously described trends of enrichment to large variant datasets and generating insights into variant enrichment in protein domains.

## Functional and proteomics features distinguish variants in health and disease

We next asked whether our statistical framework could generate new insights into the relationship between missense variants and other biological features. While databases of functional annotation and measurements of transcript/protein abundance and stability are rapidly expanding, these features have been previously underexplored in the annotation of missense variants. Here, the VES framework provides metrics which could be readily explored in terms of their association with these features.

## Disease-associated and population variants affect different functional pathways

We investigated whether variants from each dataset localise to proteins which are involved in distinct functional pathways. To do this, we performed Gene Set Enrichment Analysis (GSEA) [45] on lists of proteins ranked using their whole-protein VESs (Fig 4A) calculated for each dataset. The pathway enrichment scores were then subjected to clustering and Principal Component Analysis (PCA) (see Materials and methods). As shown in Fig 4B, variant enrichment segregates pathways into 3 clusters. Strikingly, each pathway cluster appears to have distinct characteristics (see Fig 4C–4E for the pathway terms belonging to each cluster). The cluster visualised in orange is primarily composed of terms associated with cancer, growth, and proliferation, whereas that coloured in pink contains pathways associated with splicing, transcription, translation, and metabolic terms. Pathways associated with sensory perception and the immune response are found in the "green" cluster. A handful of metabolic pathways also localise to this cluster; however, these appear to be more associated with environmental response and adaptation than those pathways found in the "pink" cluster; for example, pathways associated with the metabolism of drugs and xenobiotics are found here. For brevity, the "orange," "pink," and "green" clusters will be termed the "proliferation," "nucleotide processing," and "response" clusters, respectively, for the remainder of this text. A list of pathways assigned to each cluster is given in S7 Data.

Strikingly, this visualisation reveals that population variant datasets (gnomAD rare and common) are clearly separated from the disease-associated variants by the first principal component (PC1) (Fig 4B). Additionally, COSMIC variants are separated from ClinVar variants along the third principal component (PC3) (S5 Fig). Closer inspection of the pathway enrichment data for the top (most unique) pathways in each cluster reveals a distinction in terms of the functions that different variant datasets implicate (Fig 4C–4E). "Response" pathways appear to be enriched in variants in all 4 datasets, while "proliferation" pathways are consistently enriched in COSMIC variants alongside enrichment in ClinVar variants in a subset of these pathways. Enrichment in ClinVar variants is apparent for specific "nucleotide processing" functions, e.g., ribosomes.

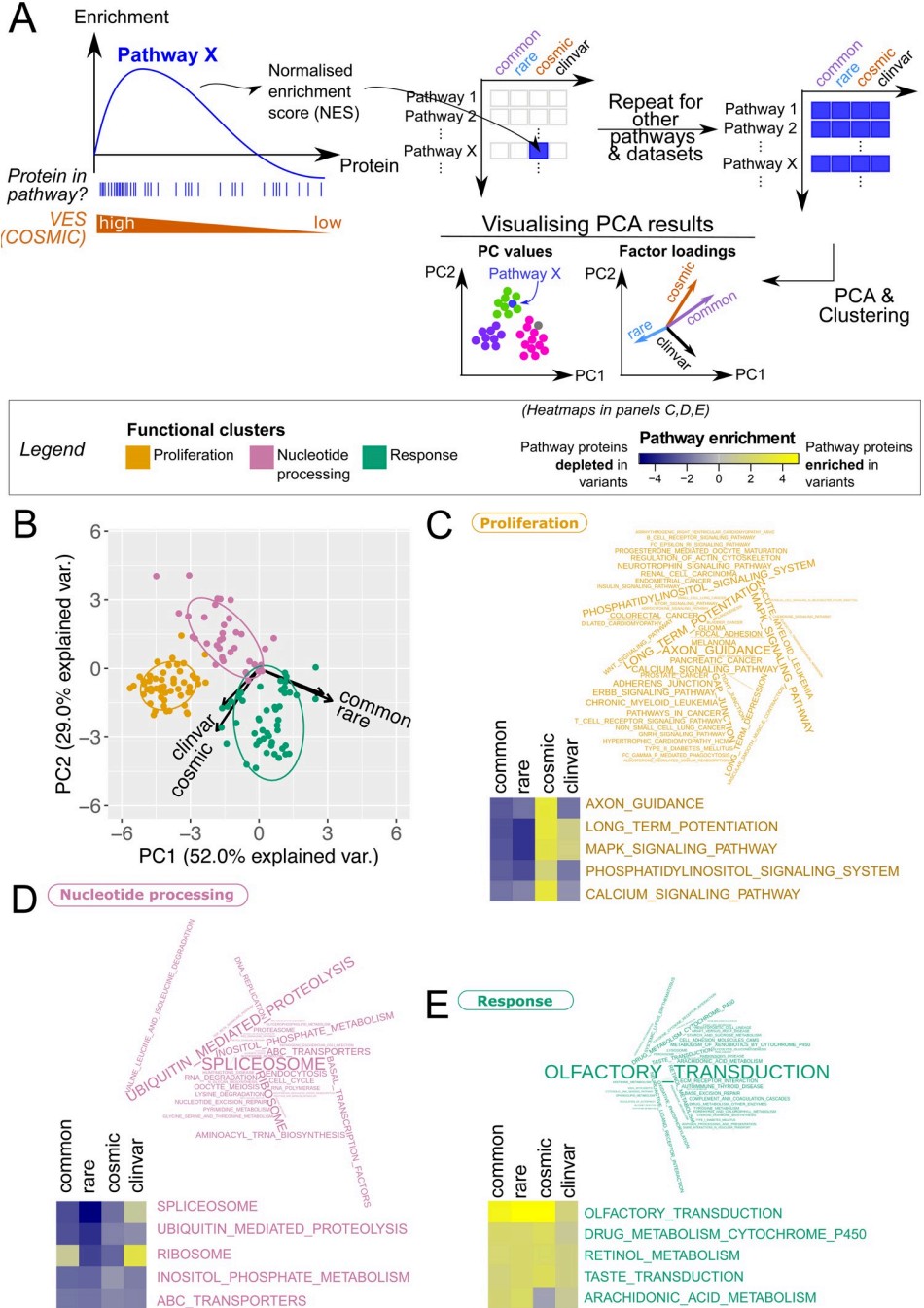

**Fig 4. Pathway clusters defined according to protein-wise variant enrichment.** (A) Schematic of GSEA. GSEA was performed over lists of proteins ranked by VES; here, enrichment over a pathway X for COSMIC variants were illustrated. The process was repeated over all KEGG pathways and the resulting NES matrix was subject to PCA and clustering analyses. (B) At the whole protein level, KEGG pathways form 3 clusters ($k$-means), here visualised as projected onto the first 2 principal components of the PCA. Pathway enrichment patterns are clearly distinct between COSMIC, ClinVar, and gnomAD (rare/common) data, as evidenced by the visualisation of factor loadings (arrows). See S5 Fig for projection onto 3 principal components. (C–E) Pathway terms visualised for the "proliferation" (C), "nucleotide processing" (D), and "response" (E) clusters, and sized by their cluster uniqueness score. The latter is defined as the average of the Euclidean distances to the two other cluster centres. For the top 5 unique pathway terms for each cluster, their pathway enrichment scores calculated with the 4 variant sets are also visualised in a heatmap. S7 Data contains the full list of KEGG terms mapped to these clusters. See S6 Data for the enrichment scores. GSEA, Gene Set Enrichment Analysis; KEGG, Kyoto Encyclopedia of Genes and Genomes; NES, normalised enrichment score; PCA, Principal Component Analysis; VES, Variant Enrichment Score.

## Proteomics and transcriptomics features are associated with variant localisation

Proteins, of course, do not function in isolation but in the crowded environment of the cell [46]. Therefore, the properties of proteins in cells, including their quantities, turnover rates, and thermal stability, can crucially affect the fitness of a protein to perform its function. Here, we ask if variant enrichments are associated with these proteomics features. We have made use of large-scale proteomics data, including protein abundance data for various organs from PaxDb [10], proteomics surveys of protein half-lives and thermal stability [14,15], together with transcriptomics data (GTEx database [9]), to explore relationships between these features and variant localisation.

We first compared the thermal stability and abundance of proteins enriched in each class of variants. This comparison demonstrates that for proteins affected by ClinVar variants, their wild-types tend to be more stable and abundant in comparison to those proteins enriched with gnomAD variants (S6 Fig). Extending to the entire proteome, the protein-wise VES of disease-associated variants displays positive correlations with protein abundance, expression, half-life, and thermal stability, whereas population variants exhibit the opposite trend (Figs 5A and 5B and S7–S9). However, zooming into the enrichment of variants in the core of protein structures, we found that in comparison to all regions of proteins with resolved structure, proteins more enriched in ClinVar variants in the core tend to be less abundant and less stable, whereas the contrary is true for rare population variants (Fig 5). Thus, our results indicate 2 competing trends for disease-associated variants: (i) disease-associated variants tend to localise to more abundant and stable proteins, which may suggest that these proteins are more sensitive to perturbation by variants; (ii) disease-associated variants in protein cores tend to localise to less stable proteins, which is consistent with the idea that such proteins might be more easily destabilised to a degree at which function is deleteriously impacted (see Discussion). gnomAD common data also show negative correlations with protein stability, for variants occurring at the core; this could potentially support the argument presented by Mahlich and colleagues [47] that common variants could affect molecular function more than rare variants. However, we believe this is more likely to be due to the fact that very few common variants localise to protein cores, as shown in Fig 2B, resulting in sparse statistics (i.e., the correlation is calculated over VESs which are already very low). Analogous correlations for variant enrichments at protein surfaces display opposite trends to those observed at the protein cores (see S7 Fig). Due to the relative sparsity of variants which map to protein interfaces, we believe it is difficult to draw robust conclusions from any trends observed for correlations of proteomics data with variant enrichment at protein–protein interaction sites.

We highlight two more observations in terms of the interplay between proteomics features. First, the various proteomics features examined here are interdependent. Protein abundance and thermal stability are significantly correlated with one another (see S11 Data), in agreement with the work of Leuenberger and colleagues [48]. Moreover, thermal stability is correlated with the density of the protein core (see S3 Text and S10 Fig), albeit with a low correlation coefficient. The correlation values displayed in Fig 5 are also typically of a weak effect. Therefore, the interplay between variant enrichment and proteomics features appear multifaceted and complex. Secondly, in the analysis of protein abundance, the trends observed with variant enrichment at both full-length proteins and specifically the protein core are less pronounced for cell line data and break down for extracellular fluids (saliva and urine, Fig 5B). The correlation is most evident for tissues containing long-lived cell-types, such as the brain, ovary, and testis. Transcriptomics data (S8 Fig) again reinforce this picture, albeit with less contrast

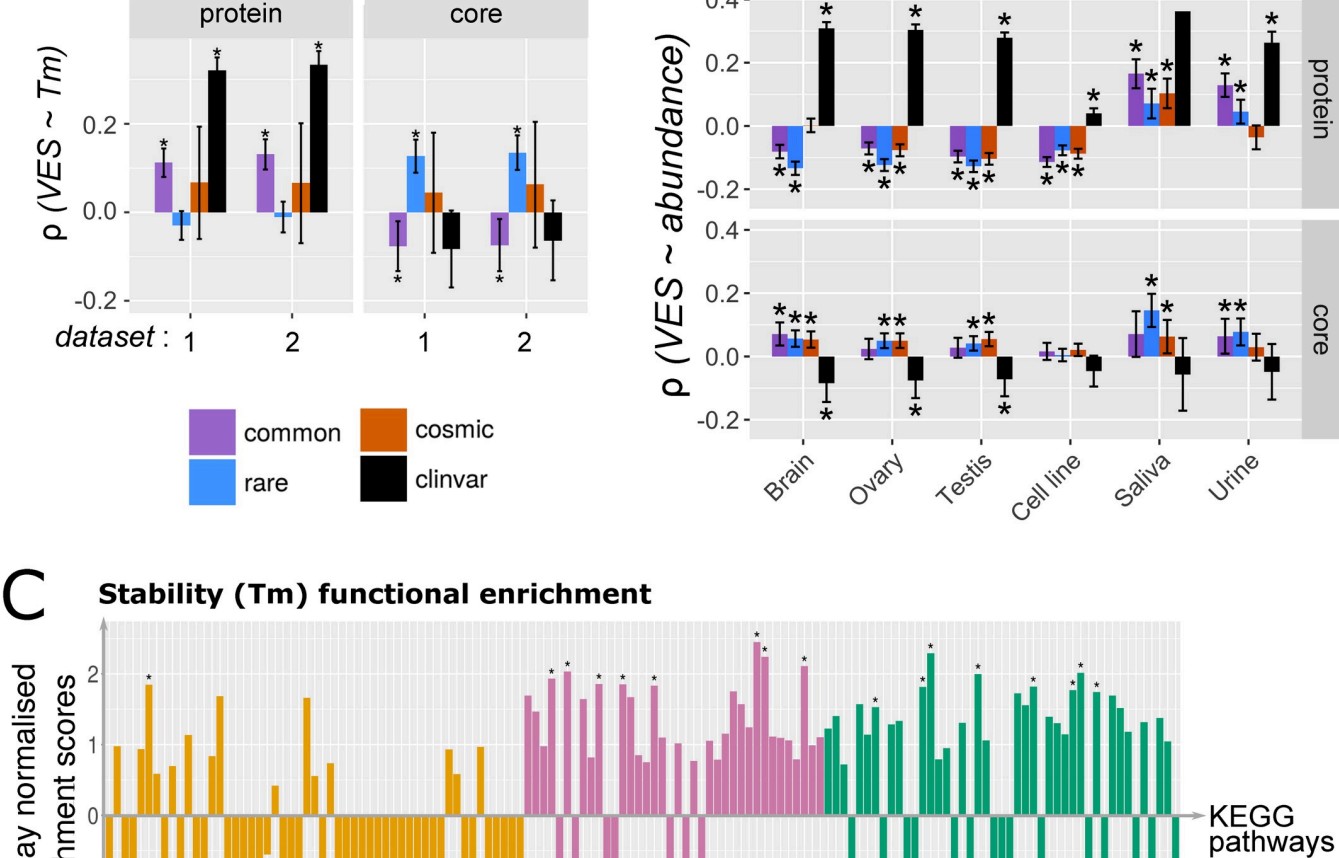

**Fig 5. The protein-wise enrichment of missense variants in comparison to protein abundance, expression, and stability.** Spearman correlations for missense variant enrichment (quantified as VESs) at the full-length protein and the core with (A) protein stability in terms of melting temperature (Tm, ˚C) and (B) protein abundance (ppm) are depicted here. For (A), the Tm data was taken from epmid26379230, in which 2 measurements of Tm in the absence of any drug treatment were available; both measurements are considered and are denoted datasets "1" and "2" in the plot. For (B), only data from selected tissue types are listed. See S7 Fig for the complete list. S9 Data contains the underlying data. (C) Functional enrichment of proteins in KEGG pathways according to Tm. The NES for each pathway is shown on the vertical axis. KEGG pathways are listed on the horizontal axis and grouped to the 3 clusters as defined in Fig 4. See S11 Fig for a complete list of pathways depicted here. S10 Data contains the underlying data. KEGG, Kyoto Encyclopedia of Genes and Genomes; NES, Normalised Enrichment Score; Tm, melting temperature; VES, Variant Enrichment Score.

between datasets (particularly at the protein core). This brings finer granularity into assessing the impact of variants in different organs and contexts.

We finally ask whether correlations with these proteomic and transcriptomic features could be associated with the specific functional roles of the involved proteins. For the majority of proteomic and transcriptomic features, no clear associations with the functional clusters identified in Fig 4 can be detected (see S11–S14 Figs). An exception to this is protein thermal stability: Pathways which belong to the "proliferation" cluster are clearly enriched in proteins of

lower stability than the other 2 clusters (Fig 5C). This suggests that proliferation-related proteins may be vulnerable to disruption by mutations which localise to their already unstable cores. Taken together, these analyses provide fine molecular details into defining both the resilience towards variants, and the sensitivity towards variants, for a given protein (see Discussion). Moreover, the association of variant enrichment with features such as abundance and stability is indicative of the condition (disease/health) associated with the variants.

## Probing the spectrum of variant pathogenicity using protein features

The analyses above extract molecular features which distinguish between ClinVar, COSMIC, and gnomAD common and rare variants. While we successfully segregate common and disease-associated variants using pathway, structural, and proteomics features, variant impact is a continuum [49] ranging from benign to very damaging towards protein function. Instead of grouping variants by the databases from which they are collected and comparing these disparate variant subsets, we therefore ask whether the protein features discovered above could distinguish variant impact across a continuous spectrum, thereby validating the utility of these features in segregating missense variants.

## Rare variants are similar to common variants

We first vary the criteria with which to define rarity of variants in the gnomAD set, to examine whether extremely rare variants would show characteristics akin to disease-associated variants. Fig 6 demonstrates that rare variants are more similar to common variants, both in terms of the functional pathways they affect and in terms of the protein regions they localise to (core, surface, and interface, order and disorder). If more stringent MAF thresholds are used to define rare variants, their properties move towards those of disease-associated variants but still remain closest to those of common variants (Figs 6 and S15). A visible separation between common and rare variants, especially in the pathway analysis, can only be seen if an extreme MAF cutoff ($<0.00001$) is used. This reinforces the boundary between population and disease-associated variants and supports the distinction in terms of molecular characteristics associated with rare population variants and disease-associated variants.

## Variant impact predictors validate structural and proteomics features

The segregation of variants into disease (ClinVar, COSMIC) and healthy (gnomAD) subsets could not capture the continuous variation in their molecular impact. We therefore seek to examine the association of the protein features we have discussed above, with orthogonal measures which treat, as a continuous variable, the impact of variants pooled from all the datasets. If such association persists, this suggests that the features described above do indeed have segregating power to identify pathogenic variants from polymorphisms which carry little impact on biological function. We pooled all analysed variants together (Fig 7A) and rank them by 2 in silico variant impact predictors, REVEL [7] and CADD [16]. Based on the scores provided by these predictors, we proceeded to compare "tolerable" and "damaging" variants labelled by these predictions in terms of the features we discussed above. The majority of ClinVar variants were classified damaging (Fig 7B), whereas the majority of COSMIC and gnomAD variants were deemed tolerable. Under our chosen cutoff (score = 20), CADD classifies a greater proportion of variants as damaging (S16 Fig). We found that damaging variants are enriched in protein cores and interacting interfaces while depleted in protein surfaces (Fig 7C), similar to the ClinVar and COSMIC variants above (Fig 2). Tolerable variants show the opposite trends which resemble that of the gnomAD variants. Similarly, protein stability and abundance also segregate tolerable from damaging variants in the same way for ClinVar/COSMIC versus

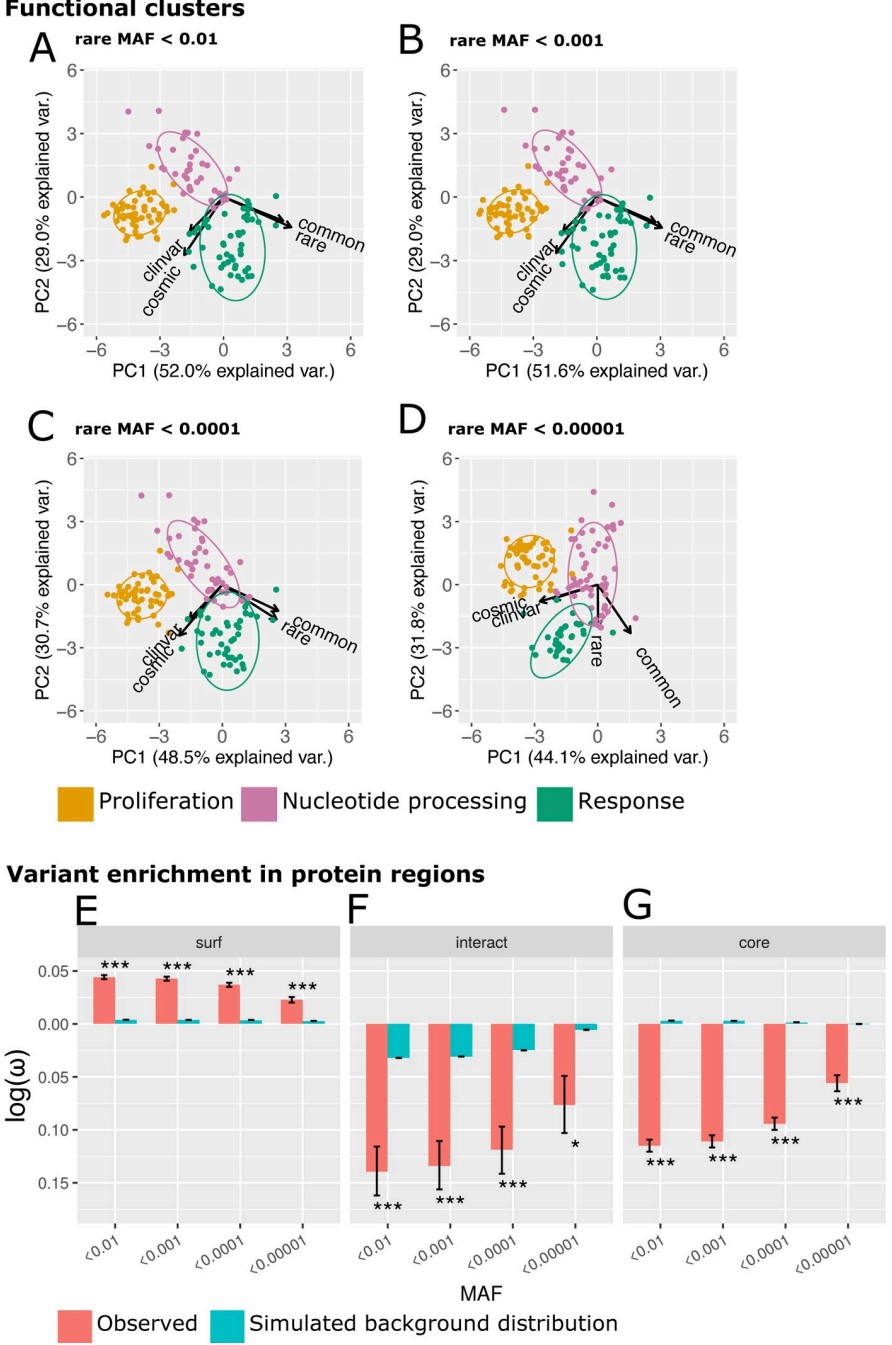

**Fig 6. Rare variants are similar to common variants.** (A–D) As for Fig 4B, but only the first 2 PCs are depicted, and, in separated panels, increasingly stringent MAFs used to define rare variants. MAF cutoffs of 0.01 (panel A, data identical to Fig 4B), 0.001 (B), 0.0001 (C), and 0.00001 (D) are considered here. See S6 Data for the underlying data. (E–G) The localisation of rare variants to protein surface (E), interface (F), and core (G). Rare variants have been defined using different MAF cutoffs as shown on the x-axes. Here, the density metrics ($\omega$) were log-transformed such that negative values indicate a depletion of missense variants, while positive values indicate enrichment. Results for the observed variants (red bars), as well as the background levels based on simulated null distributions (cyan bars), are shown. See S2 Data for the underlying data for the observed statistics. See S12 Data for statistics calculated for individual realisations of the simulated null distribution. MAF, minor allele frequency; PCs, principal components.

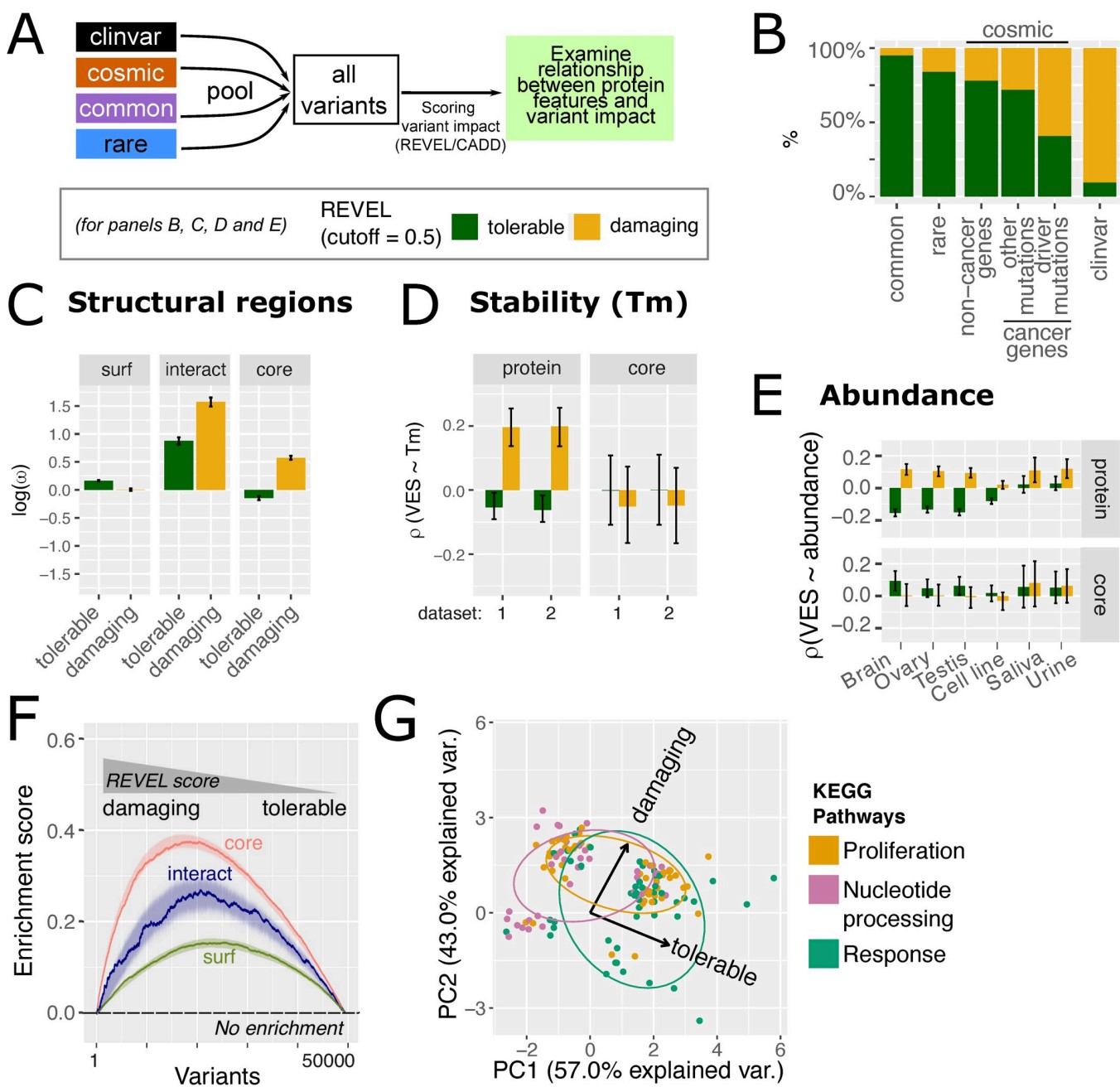

**Fig 7. Orthogonal variant impact predictions validate structural and proteomics features.** (A) Schematic of pooling variants and annotating variant impact. (B) Breakdown of variant impact classified by REVEL in the variant datasets. For COSMIC variants in cancer genes, variants were segregated depending on whether they are driver mutations (curated in IntoGen [50], version 2016.5). (C) The enrichment of tolerable and damaging variants in different protein structural regions. Variants are annotated using REVEL. The bars represented the median density (*ω*, here taken logarithm such that negative values indicate depletion and positive values indicate enrichment) of 1,000 bootstrapped samples, each a subset of 50,000 variants. The error bars represented 95% confidence intervals from such bootstrapping. See S16 Fig for analogous results for CADD. (D–E) The correlation between VES calculated at the whole-protein level ("whole") and the protein core, with protein stability (panel D, melting temperature or Tm) and abundance (E). Identical to Fig 5 but with the "tolerable" and "damaging" classification under REVEL score. See S17 and S18 Figs for data on CADD, and plots for all tissues represented in the abundance dataset. (F) The enrichment of surface, core, and interacting interface over variants ranked by REVEL score. The enrichment score from the GSEA procedure was plotted here. The absence of enrichment would result in a flat line at 0 (dashed black line). Curves represent data from 1 representative bootstrapped sample; the ribbons indicate 95% bootstrapped confidence intervals. (G) Pathway enrichment analysis for tolerable and damaging variants as defined by REVEL, i.e., underlying data identical to that of panels (B–D). Here, pathway enrichment scores were projected on 2 principal components analogous to Figs 4B and 6. Pathways were categorised using the scheme defined in Fig 4B. See S19 Fig for analogous results for CADD. S13 Data contains underlying data for all panels of this figure. GSEA, Gene Set Enrichment Analysis; Tm, melting temperature; VES, Variant Enrichment Score.

gnomAD variants (Figs 7D–7E and S17 and S18): Damaging variants tend to localise to more stable and more abundant proteins. However, such variants are more likely to be found in the core of proteins which are less stable and less abundant. These distinctions described here are also present if variants are binned using CADD scores (S17 and S18 Figs). These data indicate that the molecular features that we discovered via comparing ClinVar, COSMIC, and gnomAD variants are independent from such crude labelling based on the databases from which these variant data are collected. Rather, they are associated with variant impact, as assessed by orthogonal measures independent from our approach.

While these data serve as validation of the features we have discovered, the comparisons presented involve binning variants by their impact scores, which are based on arbitrarily determined cutoffs. To address this, we treated these impact predictions as continuous variables, ranked all variants accordingly, and repeated this analysis. By means of a GSEA approach which was also utilised above, we found that protein core and interface variants are indeed enriched in damaging variants, significantly more than the protein surface (Fig 7F). This further demonstrates the validity of structural localisation as a reliable feature to distinguish between tolerable and damaging variants. Finally, tolerable and damaging variants are also enriched indistinct functional pathways (Fig 7G). Some "response" pathways tend to be associated with tolerable variants, while "proliferation" and "nucleotide processing" pathways are more associated with damaging variants, analogous to the analysis on disparate variant sets as we see previously in Fig 4. Taken together, using orthogonal variant impact predictors, we validate that structural and proteomics features do indeed hold segregating power for tolerable and damaging variants. This suggests the use of the features discovered in our analyses above as a promising avenue to construct next-generation variant impact predictors (see Discussion).

## Discussion

Variants found in diseased and healthy populations are distributed across the proteome, each exerting a varying impact on molecular function. A detailed analysis of the patterns of variant localisation could help in understanding the functional constraints that different parts of the genome experience and improve the interpretation of variant impact. Throughout this work, we show that missense variants in the general population, considered nominally healthy, show properties distinct from those in disease cohorts, from both macroscopic ("omics" features and functional pathways) and microscopic (protein structural localisation) perspectives. Importantly, these molecular properties are not dependent on arbitrary classification into ClinVar, COSMIC, and gnomAD datasets, as we validated using orthogonal state-of-the-art variant impact predictors (Fig 7). Additionally, we find that the properties of rare variants remain close to those of common variants. These findings contrast with other observations [47] which suggest that common variants have more impact on molecular function than rare variants. In this study, only for a few proteomics properties, such as the thermal stability and abundance of the affected proteins, common variants appear closer in character to disease-associated variants than to rare variants. However, for these few properties, the results might not be robust due to sparsity of the data. Rare genetic variations are abundant across individuals [51,52], with some predicted to confer a regulatory impact [53] or loss of function [54]. Alhuzimi and colleagues [55] suggest that the properties of genes enriched in rare population variants are similar to those enriched in disease-associated variants and are thus good candidates for discovering novel disease associations. Instead, we show that proteins enriched in rare variants are, based on the associated functional pathways, most similar to those enriched in common variants (Fig 6). Moreover, our results show that population variants implicate

functions mainly associated with environmental response (Fig 4), in agreement with results from evolutionary studies reviewed in [56].

We have dissected the extent of variant enrichment in diverse datasets and across different protein regions (Fig 1). Whereas protein structural information has been utilised to annotate genetic variants and prioritise impactful variants for further investigations, many of the published methods focus on 3D-structural "hotspots," prioritising variants which cluster in three-dimensional space (e.g., in [22,37,38]). Here, we have adopted an alternative approach and quantified enrichment of missense variants without the precondition of spatial clustering. This provides an unbiased resource to map missense variants to protein structural data. The calculation of variant enrichment, as an additional layer of annotation, provides a unique link between cataloguing sequence variants and understanding both their mechanistic and functional effects. This supplies invaluable information to researchers studying specific proteins or domains, or focusing on proteins involved in a particular function (e.g., DNA binding; S3 Text). By analysing the enrichment of variants in protein regions (core, surface, interface, disorder and order, PTM vicinity), we recapitulate trends observed by previous studies (e.g., in the comparison of oncogenes and TSGs; Fig 1E) [20,21,25,26,34] but also shed light on the debate as to whether somatic cancer variants are enriched in interface regions, by simulating null distributions of variants (S3 Fig). These simulations show that it is essential to consider that variants from different datasets are not uniformly and randomly distributed throughout the proteome. A similar simulation-based approach was taken by Gress and colleagues [25], but they found no significant enrichment for COSMIC variants in interface regions. While they analysed a filtered set of mutations likely to play a driver role, we investigated all somatic variants and addressed separately mutations that localise to defined cancer and noncancer genes. Our analysis has of course been limited by the number of proteins with available structural data, despite enrichment with homologous structures. We are also still limited by the structural coverage of protein interactions; although enough data exist to uncover broad trends, our analyses of protein–protein interaction sites generally lacked statistical power. Moreover, it is likely that a more detailed picture will emerge if different classes of protein interactions (e.g., transient versus permanent interactions) could be probed systematically. We envisage that the recent advances in cryo-electron microscopy [57], and the integration of structural data derived by a variety of techniques [58], will further increase the structural coverage of the protein–protein interaction network, enabling such finer-grained analyses in the future.

Our analysis of probing the associations between missense variant enrichment and proteomic features is, to the best of our knowledge, unprecedented, and has only been made possible due to the recent release of large-scale proteomics data [10,14,15,48]. We observe correlations which suggest an interplay between variant enrichment, protein abundance, and thermal stability (Fig 5). Together with functional pathways and structural localisation, we have identified a set of features, based on which different parts of the proteome could be assessed for their tendency to be enriched in disease-associated or population variants (Fig 8). Population variants are enriched on protein surfaces but depleted in core and interacting sites, and tend to be found in less abundant, less stable proteins. These features could potentially contribute to "resilience" towards missense variants, by limiting the impact on proteins harbouring such variants. On the other hand, disease-associated variants localise preferentially to proteins which are highly expressed and abundant (Fig 8). However, when selectively looking at variants mapping to protein cores, which presumably could bring about the most dramatic impact on fitness, disease-associated variants are actually associated with cores of less stable, less abundant proteins (Fig 5). Such proteins are conceivably easier to inflict damage at the core (although it is also possible that some of these proteins are not globular but are instead more

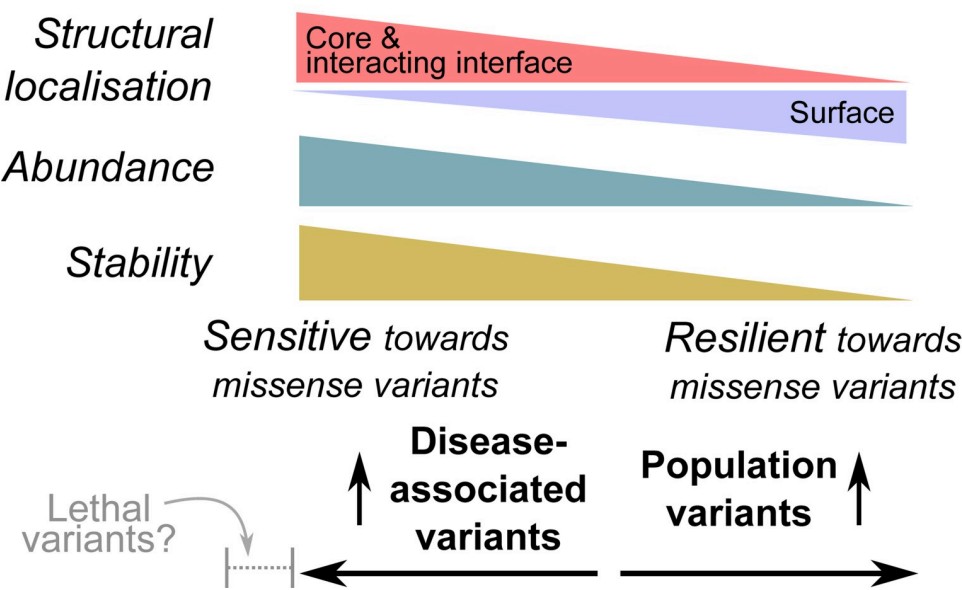

**Fig 8. Summary of analyses.** Here, we have explored patterns of structural localisation, abundance, and stability for proteins enriched in disease-associated and population variants respectively. These molecular attributes determine the resilience and sensitivity of the proteins towards missense variants (see Discussion).

extended in conformation, see below). The combination of these molecular features could also suggest the likely selection pressure a protein could experience under different contexts. For example, certain proteins, possibly further to the left of the spectrum presented in Fig 8, could show a more extreme combination of molecular features compared to those proteins discussed here to be enriched in disease-associated variants. These proteins are likely to be highly sensitive towards variants, such that any of such variants would be lethal (Fig 8) and be eliminated via selection; these lethal variants suffer from undersampling in the data analysed here. On the other hand, some variants which localise to sensitive proteins may bring benefits to cell viability; these variants could ascertain a role in driving cancers.

Our work highlights a set of rules which could explain the impact of variants. For instance, one could be fairly confident that a variant can be disruptive if it localises to the core of an abundant, stable protein. This type of variant annotation could be valuable to clinicians in interpreting variants observable in any given patient. These rules correlate with impact prediction scores offered by CADD and REVEL (Fig 7). Here, by annotating structural and proteomics features to the proteins harbouring variants, we offer explanations of the possible mechanisms through which a variant might confer its impact, enriching information provided by variant impact scores which appears intuitive yet could imply vastly different molecular underpinnings [16]. The detailed set of protein features we provide could also be harnessed for more systematic improvement of variant impact interpretation. Firstly, the analysis concerning protein stability suggests it is important to consider the baseline stability of the protein in question when assessing the impact a variant could bring. A number of algorithms have used the estimated change in protein stability upon mutation ($\Delta\Delta G$) as a proxy for variant impact [27]. Serahijos and colleagues [12] found that mutations in more stable proteins generally led to greater destabilisation ($\Delta\Delta G$ variation). Based on this observation, they suggested that proteins which have evolved to become more stable are in a state closer to their peak stability,

where any changes will result in strong destabilisation. Similarly, Pucci and Rooman [13] used temperature-dependent statistical potentials to investigate the thermal stability of the structurome (all proteins with resolved structure) and concluded that mutations in proteins which are highly thermally stable lead to a larger decrease in thermal stability, compared with those in less thermally stable proteins. We observe positive correlation between ClinVar variant enrichment and thermal stability (Fig 5A), which lends support to these conclusions. However, we also point out, by considering variants at the protein core, that a mutation in an already unstable protein is more likely to result in complete/partial unfolding under physiological conditions. This is likely to be relevant to globular proteins, whereas for other types of proteins, e.g., intrinsically disordered proteins, function will be related less to the fold and the density of the protein core. These factors should be brought into consideration when interpreting the impact of missense variants.

Secondly, we show that the combination of protein structures, functional pathways, and proteomics measurements has the potential to offer valuable mechanistic insights into the properties of variants. For example, it can be clearly seen that population variants are most enriched on protein surface (Fig 2) and take part in pathways belonging to the "proliferation" cluster (Fig 4). Such pathways also appear to be enriched in proteins with less thermal stability (Fig 5C), suggesting a possible mechanistic basis underlying the localisation of variants (variants tend to localise to the surface and avoid disrupting the core of these already unstable proteins). This indicates that the combined use of such features may aid in both improving the prediction of variant impact and in assessing the underlying molecular mechanisms.

Thirdly, our analysis highlights the tissue specificity of variant impact, in terms of the stability and abundance of the altered protein. Our association analysis (Fig 5) of variant enrichment with proteomics features complements a body of research which concludes that the rate of protein evolution correlates negatively with protein expression and abundance [59], the extent of which has been found to be tissue specific; those tissues with a high neuron density demonstrating the highest anticorrelation [60]. Consistent with this, we found the largest negative correlation for the protein-wise enrichment of rare variants, from the gnomAD dataset, with protein abundance in the brain and, interestingly, also in the ovary and testis, which both harbour long-lived germline progenitor cells (Figs 5B and S8). Purportedly, the life span of long-lived cells renders them more sensitive to (and therefore necessitate protective strategies [61] against) the toxicity of misfolded proteins. Our analysis highlights the importance of considering the underlying context, specific to the affected organ alongside with the abundance and stability levels of the affected proteins, in assessing the potential impact a missense variant could pose.

Fourthly, the wealth of data presented here could have implications in the development of therapeutic strategies. Rare population variants are known to be abundant in known drug targets, potentially modulating disease risk and drug response [62]. Here, we envisage that our domain-centric landscape of variant enrichment (Fig 3), which includes the mapping of targeted drugs, besides providing another feature for the characterisation of variants, will allow for more informed decisions in optimising therapeutic strategies. For example, targets with few population variants could be selected, to minimise differential drug response due to genetic differences between individuals. Interestingly, it has recently been shown that genetic variants in such domains (GPCRs), identified in the general population, may be associated with differential drug response between individuals [63]. By viewing variant enrichment and drug availability together, such a domain-centric landscape of variant localisation has implications useful for both understanding variant impact and motivating therapeutic design.

In conclusion, with an unbiased quantitative approach to evaluate variant enrichment, our work highlights unequivocal features, from atomistic protein structural features ("microscopic") to large-scale ("macroscopic") functional pathways and proteomics features, which

could contribute in distinguishing variant pathogenicity. We believe that mapping these annotations to missense variants will aid the interpretation of their biological impact. The ZoomVar database, which we have made available at fraternallab.kcl.ac.uk/ZoomVar, will facilitate users in the structural analysis of variants. A script is downloadable from the site to allow large-scale programmatic access to the webpage for the structural annotation of user-input variant data; we also provide in the webpage precomputed data underlying all analyses presented here. Further advancement in the structural coverage of the proteome, and the exploitation of high-throughput proteomics technologies, such as those analysed here [15,48], will ultimately offer a finer-grained picture of features which segregate variants in "health" and "disease."

## Materials and methods

See Supporting information for a more detailed description.

### Data sources

In this study, we have used variant data from ClinVar (dbSNP BUILD ID 149) [32] (for germ-line disease variants), COSMIC coding mutations (v80) [31] (for somatic cancer variants), and gnomAD exome data [30] (for population variants). Genomic positions were mapped against protein sequence data from UniProt [64] and Ensembl [65], protein structural data from the Protein Data Bank (biounit database, downloaded 28/04/2017), and protein interaction data from a large nonredundant protein–protein interaction network (UniPPIN) [66], which incorporates various interaction databases [67–71] and recent large-scale experimental studies [72,73,74]. Protein thermal stability and half-life data were obtained from separate large-scale studies [14,15]. Transcriptomic data were taken from GTEx [9], while protein abundance data (protein per million [ppm]) for each tissue/sample type were obtained from PaxDb [10]. Both PaxDb and GTEx normalised data were taken directly for use without additional filtering steps.

### ZoomVar database

ZoomVar was constructed by mapping human protein sequences to resolved structures/homologues from the PDB using BLAST [75]. Protein domains were defined by scanning UniProt sequences against the PFAM seed library [76] using HMMER [77]. Per-residue mappings were performed by the alignment software T-COFFEE [78] or Stretcher [79] (which was used to map UniProt and Ensembl sequences which were not of the same length and were too long to align using T-COFFEE). These generated correspondences between PDB structures and those proteins/domains with structural coverage. Interaction complexes were inferred from homologues (defined using BLAST). As an example, if protein *A* and *B* are annotated as interacting in UniPPIN, and their structure homologues *A'* and *B'* are located in a resolved structural complex (and at least 1 residue from each protein is involved in a shared interface), residues from *A* and *B* are mapped onto *A'* and *B'* to infer their interaction interface. The partner-specific regression formula from HomPPI [80] was used to assign a score and confidence level to way. Only heterocomplexes are considered.

### Mapping of variant data

### Definition of regions

**Structural regions.**   We partitioned protein/domain into surface, core, and interface regions. Interface regions were considered to be composed of residues which bind to at least 1 protein interaction partner. The interfaces were assigned using POPSCOMP [81]. Residues with a change in solvent accessible surface area [SASA] > 15 $Å^2$ were annotated as interface

residues [82]. For surface and core regions, these were classified by considering the quotient SASA [Q(SASA)] per residue, which was computed using POPS [83]. Core residues were defined as those with a Q(SASA) < 0.15 [82]. Surface residues were defined as those with a Q (SASA) ≥ 0.15 which do not take part in protein–protein interaction interfaces.

**Order and disorder.** Disordered protein regions were predicted using DISOPRED3 [84]. We overlaid these definitions of ordered and disordered regions with Pfam domain boundaries and partitioned protein sequences into intra-domain ordered, intra-domain disordered, and inter-domain disordered regions.

**Functional sites.** Posttranslational modification (PTM sites, specifically ubiquitination and phosphorylation sites, were obtained from PhosphoSitePlus [85]. Regions close to phosphorylation and ubiquitination sites were defined as those within 8 Å in Euclidean distance, following studies (e.g., [34]) using this threshold to define regions close to PTMs.

## Classification of variant pathogenicity

Variants in each dataset were annotated according to protein region localisation using the ZoomVar database. Table 1 listed the total number of missense variants in each dataset which have been mapped to each region considered in this study.

In the exploration of variant enrichment in different structural regions, the COSMIC data was divided into "cancer genes" and "noncancer genes" subsets, taking "cancer genes" as variants in the genes which comprise both tier 1 and tier 2 of the Cancer Gene Census (CGC) (COSMIC v84). The noncancer gene subset contains all other variants. In addition, to address the effect of labelling variants by the databases from which these data are collected, variants are also pooled and annotated using 2 existing in silico variant impact predictors CADD [16] and REVEL [7] predictions, using the Ensembl Variant Effect Predictor (VEP) [86]. Variants were labelled "damaging" and "tolerable" using cutoffs discussed in the publications [7,16] of these predictors (CADD: > 20 as "damaging"; REVEL: >0.5 as "damaging"; "tolerable" otherwise). The association between such variant pathogenicity measures and the molecular features extracted when comparing the disparate variant sets were subsequently investigated. Since these predictors are independent (i.e., not constructed directly with features investigated here) from our study, this mediates the impact of arbitrary grouping variants into ClinVar, COSMIC, and gnomAD common and rare subsets, and serves to validate the segregating power of our molecular features on variant pathogenicity.

## Missense variant enrichment across levels of protein anatomy

### The protein anatomy

In this study, missense variant enrichment was quantified across the "protein anatomy," in which we partition the human proteome in different ways (see Fig 1A and 1B). We first define a list of "levels" of the anatomy, namely: (i) individual "proteins"; (ii) specific "domains" of proteins; and (iii) instances of a "domain-type" across the human proteome. See Results for detailed examples. Here, missense variant enrichment quantification was considered in both of the following scenarios: (i) for a given full-length instance of a "level," relative to all other instances at the given "level" (e.g., for epidermal growth factor receptor [EGFR] relative to all other proteins in the human proteome), or (ii) for a given "region," relative to all regions defined under a given criteria at a relevant level (e.g., for the protein core relative to all protein structural regions). For the sake of clarity, in this Materials and methods section, the instance of interest is hereafter referred to as the "entity" of interest; its use is clarified in the next paragraph and illustrated in Fig 9.

## Calculation of missense variant enrichment

To calculate the enrichment of missense variant, we first sought for a probabilistic calculation of the likelihood of observing the given number of missense variants inside an entity. We modelled this problem as the assignment of $n$ variants into various entities and calculated the probability that $k$ of those localise to the entity of interest. Assuming no bias towards any entity, the expected probability to assign 1 variant to an entity would be the ratio between the size (in terms of number of amino acids) of that entity to the total size of all entities.

This can be formulated as a classical binomial distribution problem. The binomial cumulative distributive function (Eq 1, also illustrated in Fig 9A) was used to assess the missense variant enrichment of a given *entity*, and the two-tailed binomial test was used to assess the significance of enrichment/depletion.

$$P(X_{entity} \leq k) = \sum_{i=0}^{k} \binom{n}{i} p^i (1-p)^{n-i} \tag{1}$$

where $k$ is the number of observed missense variants which localise to the given entity, $n$ is the total number of missense variants which localise to all relevant entities, and $p$ is the expected probability to assign 1 variant to the entity. $p$ is given by the following expression:

$$p = \frac{size\ of\ entity\ of\ interest}{total\ size\ of\ all\ entities}$$

As an example, when evaluating enrichment in the core of a given protein with complete structural coverage (i.e., all residues within the protein can be assigned to any one of surface, core, or interacting interface):

$$p = \frac{number\ of\ residues\ in\ the\ core}{total\ number\ of\ residues\ in\ the\ protein}$$

The definition of $n$, $k$, and $p$ is illustrated in Fig 9B and 9C.

In Eq 1, $P(X_{entity}) \leq k)$ is the cumulative probability of observing $k$ missense variants in the chosen entity. We took the approach used in [17] to identify protein interfaces enriched in cancer variants and generalised this to look not only at interfaces but other regions and levels defined in our protein anatomy (see above). Note that this formulation of the binomial model tests for the concentration (or avoidance) of variants to a specific entity, instead of modelling each "trial" of the binomial process being the generation of a variant. A detailed comparison and illustration of methods to calculate variant enrichment can be found in S1 Text.

Hereafter, we refer to the binomial cumulative distributive function of this binomial distribution as the Variant Enrichment Score (VES). Fig 9C contains specific examples of applying this scheme of VES calculation to different levels and regions of the protein anatomy.

For each analysis at the protein or domain level, the background proteome is defined as the summation of all UniProt proteins/domains containing missense variants in any of the datasets analysed. Proteins belonging to immunoglobulin and T cell receptor gene family products were filtered from all analyses (HGNC definition [87]), to avoid the inclusion of variants which could have arisen from the process of affinity maturation. For all calculations of enrichment and simulations involving protein or domain "regions" (e.g., core, surface, and interface), cases where the region is of size 0, or where the protein/domain contains no missense variants, were omitted in this analysis. Note that this framework of variant enrichment quantification, in contrary to others [22,37,38], is not designed to detect mutational "hotspots" clustered in sequence or structure space. Instead, it quantifies the extent to which missense variants are populated in the entity concerned, evaluating whether the number of such variants

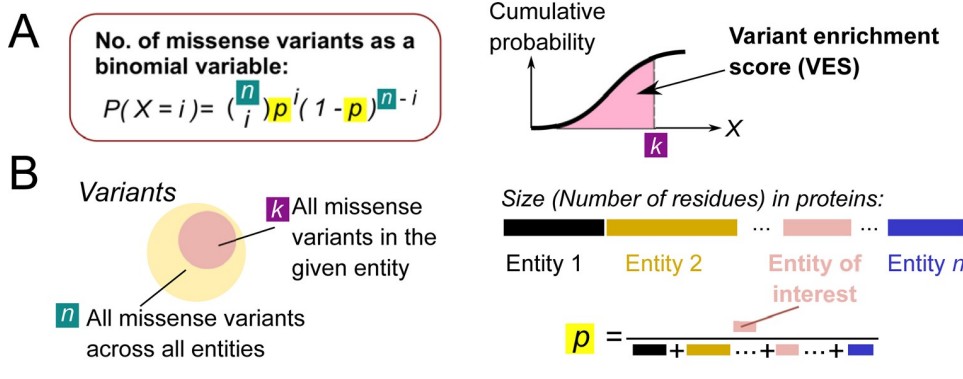

| Problem | "Entity" | Total | $p$ in VES formula |
|---|---|---|---|
| Variant enrichment in Protein X | Protein X | All proteins | $\frac{\text{Size of protein X}}{\text{Size of all proteins}}$ |
| Variant enrichment in the core of Protein X | Core of Protein X | Protein X | $\frac{\text{Size of Core in X}}{\text{Size of protein X}}$ |
| Variant enrichment in Domain Y | Domain Y | All domains | $\frac{\text{Size of domain Y}}{\text{Size of all domains}}$ |
| Variant enrichment in core of Domain Y | Core of Domain Y | Domain Y | $\frac{\text{Size of core in Y}}{\text{Size of domain Y}}$ |

**Fig 9. Illustration of missense variant enrichment calculation.** (A) The number of missense variants is modelled as a binomial variable. The cumulative distributive function of this binomial variable is taken as a VES for the level examined. (B) Illustration of the choice of parameter in defining the binomial variable used in calculating the VES. (C) Examples of defining the parameter $p$ in quantifying variant enrichment for different cases. VES, Variant Enrichment Score.

are more or less than expected. S1 Fig shows the distributions of whole-protein VES and their associated $p$-values for all the variant sets examined in this work.

The overall missense variant enrichment for each dataset was also calculated using a density-based metric $\omega$ (see Eq 2).

$$\omega(X_{entity}) = \frac{X_{entity}/S_{entity}}{X_{all\_entities}/S_{all\_entities}} \tag{2}$$

where $S$ refers to the size in terms of the number of residues, and $X$ refers to the number of variants.

This expression therefore examines the ratio between the number of variants in the entity of interest versus that of other entities, normalised by their sizes. As expected, the density shows good correlation with VES (S2 Fig); for simplicity and consistency, we use the VES metric for variant enrichment/depletion within a given entity (e.g., core of protein Z), whereas the density metric is used to describe a collection of entities at a given level/region (e.g., cores of every human protein).

Here, 95% confidence intervals were estimated via bootstrapping (10,000 iterations). The 2-tailed significance of enrichment/depletion was estimated by simulation of the null background. A total of 10,000 simulations were carried out for each dataset, in which the number of variants which localise to a given entity was kept constant, but their location within the

entity randomised. The density of variants was calculated for each simulation and compared to the actual value in order to derive a *p*-value. Simulations were performed in this way, keeping the observed number of missense variants fixed, in order to overcome bias which stems from the assumption that variants are uniformly distributed throughout the proteome.

## Enrichment analysis of gene sets

### Gene set enrichment

Gene enrichment analyses were performed using Gene Set Enrichment Analysis (GSEA), using the implementation provided by the R fgsea package [88]. Given an enrichment statistic for each query gene, the GSEA algorithm outputs a score per gene set, which quantifies the enrichment of query genes in the sets examined. This is then normalised by the size of the gene set, to give a normalised enrichment score (NES) [45]. We utilised the centred VES enrichment statistic, i.e., subtracting 0.5 from the VES, as input into the GSEA algorithm. Thus, proteins with fewer missense variants than expected would have a negative score.

### Definition of pathway clusters

The pathway normalised enrichment scores (NESs), calculated at the whole protein level for each dataset, were used to perform K-means clustering of KEGG pathways [89]. The R package NbClust [90] was used to determine the optimum number of clusters.

## Analysis of expression, abundance, and stability data

Spearman correlations of protein-wise and region missense variant enrichments with expression levels (RPKM), abundance (ppm), half-life (hours), thermal stability (Tm, in ˚C), and density (mean contacts of core C$\alpha$s) were calculated. Additionally, gene set enrichment analysis was performed as detailed above, except that the mean value for each quantity of interest was subtracted to obtain values centred around 0, allowing both pathway enrichment and depletion to be assessed (see S2 Text).

## Statistics and data visualisation

The majority of data analyses were performed in the R statistical programming environment. All corrections for multiple testing have been done using the Benjamini–Hochberg method in R (p.adjust function). Bootstrapping was performed using the boot package [91]. Spearman correlations were performed using the SpearmanRho function of the DescTools package [92]. Heatmaps were produced with either the heatmap.2 function in the gplots package [93] or the ComplexHeatmap package [94], in which clustering, wherever shown, was performed with hierarchical clustering (hclust function) using default parameters unless otherwise stated. Circos plots were generated with the Circos package [95]. Additionally, binomial cumulative distributive functions were calculated and two-tailed binomial tests performed using the NumPy package in Python [96].

## Supporting information

**S1 Text. Note on probabilistic methods to model variant enrichment.**
(PDF)

**S2 Text. Supplementary methods.**
(PDF)

**S3 Text. Supplementary results.**
(PDF)

**S1 Fig. Distribution of VES and associated *p*-values.**
(PDF)

**S2 Fig. Comparison of VES and density metrics.**
(PDF)

**S3 Fig. The density of mutations in different protein regions.**
(PDF)

**S4 Fig. The landscape of variant enrichment over a list of domain-types most enriched in pathogenic and population variants.**
(PDF)

**S5 Fig. Principal Component Analysis (PCA) of pathway enrichment of protein-wise VES.**
(PDF)

**S6 Fig. Stability and abundance of proteins enriched in variants in each dataset.**
(PDF)

**S7 Fig. Spearman correlations for missense variant enrichment, protein stability, and abundance.**
(PDF)

**S8 Fig. The enrichment of missense variants in comparison to protein abundance and expression.**
(PDF)

**S9 Fig. The Spearman correlation of the enrichment of missense variants with protein half-lives.**
(PDF)

**S10 Fig. The Spearman correlation of the enrichment of missense variants in protein cores with protein core density.**
(PDF)

**S11 Fig. Functional enrichment of proteins in KEGG pathways according to thermal stability.**
(PDF)

**S12 Fig. Functional enrichment of proteins in KEGG pathways according to protein abundance.**
(PDF)

**S13 Fig. Functional enrichment of proteins in KEGG pathways according to transcript expression.**
(PDF)

**S14 Fig. Functional enrichment of proteins in KEGG pathways according to protein half-life.**
(PDF)

**S15 Fig. The density of rare mutations from the gnomAD data in different protein regions.**
(PDF)

**S16 Fig. The distribution and density in protein structural regions of variants classified by CADD.**
(PDF)

**S17 Fig. The association between VES and protein stability for variants scored by CADD and REVEL.**
(PDF)

**S18 Fig. The association between VES and protein abundance for variants scored by CADD and REVEL.**
(PDF)

**S19 Fig. Pathway enrichment analysis for tolerable and damaging variants as defined by CADD.**
(PDF)

**S1 Data. Details of the number of missense variants which localise to different protein regions in the gnomAD common and rare, COSMIC, and ClinVar datasets.**
(ZIP)

**S2 Data. Variant density and VES of protein regions and domains.** GSEA of structural/disorder regions VES. (Data underlying Figs 2, 3 and 6E–6G and Fig B–E in S3 Text and S1–S4, S10 and S15 Figs).
(XLSX)

**S3 Data. Density of variants in protein regions in simulations of random variants.** (Data underlying S3 Fig).
(XLSX)

**S4 Data. Proteins enriched in COSMIC noncancer gene variants at protein–protein interaction sites.**
(CSV)

**S5 Data. Statistics for comparisons of structural network features between datasets.** (Data underlying Fig A in S3 Text).
(CSV)

**S6 Data. Gene Set Enrichment Analysis (GSEA) of protein VES calculated on ClinVar/COSMIC/common/rare variant sets.** (Data underlying Figs 4 and 6A–6D and S5).
(XLSX)

**S7 Data. A list of pathways annotated by functional cluster ("proliferation," "nucleotide processing," and "response").** (Data underlying Fig 4).
(CSV)

**S8 Data. Numbers of proteins and missense variants which underlie correlations between proteomic/transcriptomic features and variant enrichment.**
(ZIP)

**S9 Data. Proteomics features and correlation with variant enrichment scores.** (Data underlying Figs 5A and 5B and S6–S9).
(XLSX)

**S10 Data. GSEA of proteomics and transcriptomic measures.** (Data underlying Figs 5C and S11–S14).
(XLSX)

**S11 Data. Pairwise Spearman correlations between all studied proteomics and transcriptomics features.**
(ZIP)

**S12 Data. Density of rare variants under different MAF cutoffs, in simulations of random variants.** (Data underlying Figs 6E–6G and S15).
(XLSX)

**S13 Data. Comparison of variant enrichment and associations of damaging and tolerable variants as annotated by orthogonal impact predictors.** (Data underlying Figs 7 and S16–S19).
(ZIP)

**S1 Appendix. Detailed description of data files.**
(PDF)

## Acknowledgments

We thank Dr Jens Kleinjung for his critical reading of and comments on this manuscript, and members of the Fraternali group for valuable discussion.

## Author Contributions

**Conceptualization:** Anna Laddach, Joseph Chi Fung Ng, Franca Fraternali.

**Data curation:** Anna Laddach, Joseph Chi Fung Ng.

**Formal analysis:** Anna Laddach, Joseph Chi Fung Ng.

**Funding acquisition:** Franca Fraternali.

**Investigation:** Anna Laddach, Joseph Chi Fung Ng.

**Methodology:** Anna Laddach, Joseph Chi Fung Ng, Franca Fraternali.

**Project administration:** Franca Fraternali.

**Resources:** Franca Fraternali.

**Software:** Anna Laddach, Joseph Chi Fung Ng.

**Supervision:** Franca Fraternali.

**Validation:** Anna Laddach, Joseph Chi Fung Ng.

**Visualization:** Anna Laddach, Joseph Chi Fung Ng.

**Writing – original draft:** Anna Laddach, Joseph Chi Fung Ng, Franca Fraternali.

**Writing – review & editing:** Anna Laddach, Joseph Chi Fung Ng, Franca Fraternali.

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
