## [Editor Report · Decision Letter 0]

10 Jun 2020

Dear Dr Fraternali, 

Thank you for submitting your manuscript entitled "Missense variants in health and disease affect distinct functional pathways and proteomics features" for consideration as a Research Article by PLOS Biology.

Your manuscript has now been evaluated by the PLOS Biology editorial staff as well as by an academic editor with relevant expertise and I am writing to let you know that we would like to send your submission out for external peer review, but we would like to consider it as a Methods and Resources manuscript. If you agree with this, please remember to change the type of manuscript when you proceed with the full submission (see below).

Before we can send your manuscript to reviewers, we need you to complete your submission by providing the metadata that is required for full assessment. To this end, please login to Editorial Manager where you will find the paper in the 'Submissions Needing Revisions' folder on your homepage. Please click 'Revise Submission' from the Action Links and complete all additional questions in the submission questionnaire.

Please re-submit your manuscript within two working days, i.e. by Jun 12 2020 11:59PM.

Kind regards,

Ines

--

Ines Alvarez-Garcia, PhD

Senior Editor

PLOS Biology

Carlyle House, Carlyle Road

Cambridge, CB4 3DN

+44 1223–442810

---

## [Decision Letter · Decision Letter 1]

12 Sep 2020

Dear Franca,

Thank you very much for submitting your manuscript "Missense variants in health and disease affect distinct functional pathways and proteomics features" for consideration as a Methods and Resources at PLOS Biology. Thank you also for your patience as we completed our editorial process, and please accept my sincere apologies for the delay in providing you with our decision. Your manuscript has been evaluated by the PLOS Biology editors, an Academic Editor with relevant expertise, and by three independent reviewers.

As you will see, the reviewers find the method interesting and important for the field. However, both Reviewer 1 and 2 raise several methodological issues with the analysis and conclusions that would need to be addressed before we can consider this manuscript for publication.

In light of the reviews (attached below), we will not be able to accept the current version of the manuscript, but we would welcome re-submission of a revised version that takes into account the reviewers' comments. We cannot make any decision about publication until we have seen the revised manuscript and your response to the reviewers' comments. Your revised manuscript is also likely to be sent for further evaluation by the reviewers.

We expect to receive your revised manuscript within 3 months. 

**IMPORTANT - SUBMITTING YOUR REVISION**

*Re-submission Checklist*

*Published Peer Review*

*PLOS Data Policy*

*Blot and Gel Data Policy*

Sincerely,

Ines

--

Ines Alvarez-Garcia, PhD,

Senior Editor,

ialvarez-garcia@plos.org,

PLOS Biology

Reviewers’ comments

Rev. 1:

Laddach, Ng et al. attempt to quantify variant enrichment in human proteins to decipher basic principles underlying the differences between pathogenic and benign variants. This is a very important topic of study and its impact to disease researchers is high. The author's develop a measure, VES, of determining enrichment of variants across several levels of protein hierarchy (i.e. entire protein, domains and domain-type). The author's also incorporate proteomic features such as a protein half-life into their analysis providing a unique evaluation that is much needed in the field. Unfortunately, I have deep concerns about the underlying statistics and ultimately the interpretability of the VES measure which impact much of manuscript. Due to these concerns, the conclusions of the paper are all in question.

Major:

The author's develop a Variant Enrichment Score (VES) to "… assess the missense variant enrichment of a given entity…". Their formulation of the binomial test in computing the VES is incorrect. The binomial test requires definitions of k, n and p. To test enrichment in a given entity, k should equal the number of missense variants in the entity (as the author's properly define), n should be the size of the entity and p should be the expected probability of missense variants in the entity. The author's define n as "the total number of missense variants which localise to all relevant entities" and p as "the ratio of the  size of the entity of interest (in terms of number of residues) to the total size of all relevant entities". It is unclear how these definitions of n and p substitute for the expected definitions and likely result in uninterpretable results.

The manuscript is lacking validation of the Variant Enrichment Score. A global analysis of how the VES behaves on known enriched examples (positive controls) as well as known depleted examples (negative controls) would be beneficial for evaluation. Individual positive control examples for each entity type (ie. protein, domain and domain type) would also aid the reader in how to interpret the VES.

Figure 6A shows P53 full length domain-type with a VES score of being depleted in gnomad_common. My expectation would be that at least one of the three individual "domain-type structural regions" (i.e. surf, interact or core) in figure 6B would show depletion for gnomad_common and none would show enrichment. Figure 6B shows all three domain-types as enriched. Can the author's explain the inconsistency?

Minor:

Figure 3 legend refers to "red arrows" that are absent from figure.

Rev. 2:

In this interesting work by Laddach, Ng, and Fraternali, the authors enumerate a series of protein-centric features and combine these with available genetic variation data from COSMIC, ClinVar and gNOMAD resources to create a data resource (ZoomVar), and to compute variant enrichments relative to these features. Enrichments are calculated using a binomial test and a density-based metric. The authors report some findings consistent with previously reported mechanisms for rare disorders and oncogenesis. While many aspects of the presented research are appealing (the examination of protein interfaces, disordered regions, etc.) the overall presentation and message of the manuscript is very difficult to follow as it is currently structured. For example, the authors state in their conclusion that their work highlights the complex interplay between different factors that influence pathogenicity - this was not in doubt given the prior literature. They then state that "these insights" will be useful for variant prioritization, but it is unclear what specific insights they are referring to. I am entirely sympathetic to the tremendous effort in compiling these data, but I would suggest refocusing the manuscript to emphasize fewer, more concise points you have identified from your analyses. Specific comments and points of confusion are the following:

1. The abstract and much of the introduction prepares the reader for evaluations of variant pathogenicity, yet the authors do not propose any specific method for ranking variant pathogenicity. I think this is OK, but I would refocus the manuscript away from predicting variant pathogenicity and more toward understanding the critical features of proteins that induce phenotypic changes rather than focusing on the variants themselves.

2. The introduction is a bit broad in scope and doesn't provide an understanding of what this paper adds to the scientific literature, or at best does so in vague terms. Is there a specific finding from this paper that the authors wish to highlight? The importance of ordered versus disordered regions, etc?

3. The authors state that 3D structure-based evaluation has not been performed on population or Mendelian disease variants, but several other studies the authors cite seem to have done this - 15, 16, 56.

4. The methods section could be expanded to include any filtering or normalization steps used on the variant sets, GTEx transcripts and PaxDB data. If no filtering was performing, it would be useful to explicitly state this.

5. In general, the rationale and interpretation of the VES is not clear, and some elaboration on the interpretation, particularly with respect to multimers and the potential violations of assumptions surrounding protein size should be discussed. A distribution of the VES scores for each dataset would be useful to present, and it could be useful to generate a composite distribution of VES across all three datasets as a null distribution.

6. The PC plots are somewhat difficult to follow. In figure 3A, I presume the authors ran the VES across all proteins for each of the four variant sets, then ran PCA on the GSEA scores for each of those sets? Each data point represents a pathway within the enrichment space? A detailed caption enumerating the steps toward this figure would be helpful.

7. Figures 4 and 5 seem to contradict one another. Looking at figure 4 there appears to be no enrichment or depletion in the surface or core features when you look at the pathway level (presumably scored by VES?), but in figure 5 when you use a different density metric at the protein level, the authors claim there is evidence of this - how do you describe this discrepancy? It is also unclear why the authors are using both metrics (VES and density) instead of focusing on one, or integrating the two more fully. It would also be helpful to know the correltation between the VES and density-based metric.

8. Figure captions should be improved to match the figures - several colors and labels are mis-specified, and some inline references to figures point to the wrong ones (line 587).

9. For the population-level variation, the interpretation of enrichments is difficult to follow - these are proteins that are susceptible to natural variation? Would the opposite enrichment not be more interesting - protein regions that are devoid of variation?

10. What is the sensitivity of the enrichment test for extremely low counts? It seems likely that some proteins may harbor only one or two variants from some of these sets - especially for sub-regions of the protein.

11. While the comparisons to CADD and REVEL are encouraging as a positive control, doesn't this result also imply that these scores are classifying pathogenicity well without using protein-structure information? Furthermore, it seems likely that these scores were trained in part from clinvar variants - what does this analysis provide that comparisons to Clinvar do not?

12. The emphasis on pathway-related figures and content seems to distract from the more biochemical points of the manuscript. The effect of a variant on the biochemical properties of a protein is already complex and difficult to explain - this is compounded by the pathway-level analyses, and other more interesting associations (like the PTM and VES) are more interesting from a protein perspective, but this is relegated to supplementary figures.

I would like to re-emphasize that the work presented is important and interesting and does make several points worth contributing to the literature as a resource - but the current presentation of the results doesn't quite do it justice, and leaves the reader unclear how to use such a resource in future work.

Rev. 3: Rob Ewing – note that this reviewer has waived anonymity

Overview

Laddach et al have performed a comprehensive analysis of missense variants. This analysis focuses on comparison of missense variants in proteins associated with disease (cancer), germline disease and variants present in the wider population. An important motivation of the study is to understand missense variants and the effects they have so as to better be able to distinguish variants that are associated with disease. The paper consists of detailed analyses of protein missense variants across the 3 contexts and release of a database resource.

The study identifies some interesting correlations and findings which will be of general interest. In addition, this paper is anice example of how integration of different data can be used to ask important and wide ranging biological questions. The authors indicate that this is the first example of using proteomics data as measurements of stability and abundance to explore how missense variants are associated with these properties. As a PLoS Biology resource, the paper, associated data and the database will be of immediate use to others. The work is also very well explained. In particular the authors have made good efforts to explain the computational techniques used in a clear graphic form. This will aid understanding and accessibility in particular for non-computational biologists.

Specific Comments

1. Clarifying the variant enrichment calculation. Please indicate whether the method differs from previously published. Also clarify or explain - the model assumes that each amino-acid in each protein has an equal probability of accumulating a missense variant? Also are the entities in Figure 1 only proteins or also protein domains? If so - is there normalization for the length of the domains as there is for the length in AA of each protein?

2. It was not clear exactly how GSEA was used in the context of variant enrichment. Were proteins ranked according to VES? Should 'substrating' read 'subtracting'? (line 184)

3. Figure 7B is too small - in particular because there are several significant results shown it should be larger.

4. Is there a previous rationale for classifying regions close to PTMs if within 8 angstroms? (Line 113)

5. Add more explanation for Equation 2 .. what X is etc

---

## [Decision Letter · Decision Letter 2]

12 Feb 2021

Dear Franca,

Thank you very much for submitting a revised version of your manuscript entitled "Missense variants in health and disease affect distinct functional pathways and proteomics features" for consideration as a Methods and Resources at PLOS Biology. Please accept again my apologies for the delay in providing you with our decision. This revised version of your manuscript has been evaluated by the PLOS Biology editors, the Academic Editor and the three original reviewers.

The reviews are attached below. You will see that the reviewers find the new version of the manuscript very much improved, nevertheless Reviewer 1 raises a couple of points that need clarification and Reviewer 2 thinks the manuscript would benefit from some changes to the structure and section titles to improve accessibility for our broad audience. In addition, we think the abstract and title can also improve. Please rewrite the abstract to make it clearer and consider this suggestion to improve the title:

"Pathogenic missense protein variants affect different functional pathways and proteomic features than variants not linked to disease"

We are pleased to offer you the opportunity to address the points raised by the reviewers in a revised version that we anticipate should not take you very long. We will then assess your revised manuscript and your response to the reviewers' comments and we may consult the reviewers again.

We expect to receive your revised manuscript within 1 month.

**IMPORTANT - SUBMITTING YOUR REVISION**

3. Resubmission Checklist

a) *Published Peer Review*

b) *PLOS Data Policy*

Please provide the data underlying the following figures, and make sure you mention in the corresponding figure legends WHERE THE DATA CAN BE FOUND:

Fig. 3B-E; Fig. 4A-F; Fig. 5C; Fig. 6A-C; Fig. 7A-G; Fig. 8B-G; Fig. S3; Fig. S4A-C; Fig. S5A-F; Fig. S6; Fig. S7; Fig. S8; Fig. S9; Fig. S10; Fig. S11; Fig. S12; Fig. S13; Fig. S14; Fig. S15; Fig. S16; Fig. S17; Fig. S18; Fig. S19; Fig. S20; Fig. S21; Fig. S22; Fig. S23; Fig. S24; Fig. S25; Fig. S26 and Fig. S27

d) *Blurb*

Please also provide a blurb which (if accepted) will be included in our weekly and monthly Electronic Table of Contents, sent out to readers of PLOS Biology, and may be used to promote your article in social media. The blurb should be about 30-40 words long and is subject to editorial changes. It should, without exaggeration, entice people to read your manuscript. It should not be redundant with the title and should not contain acronyms or abbreviations. For examples, view our author guidelines: https://journals.plos.org/plosbiology/s/revising-your-manuscript#loc-blurb

Best wishes,

Ines

--

Ines Alvarez-Garcia, PhD,

Senior Editor,

PLOS Biology

Reviewers’ comments

Rev. 1:

The authors of Laddach, Ng et al. have addressed most of my major concerns. In particular, their supplemental note explaining the formulation of their Variant Enrichment Score (VES) greatly helps in the understanding and ultimately the interpretation of the score. I do still believe the main text describing the VES calculation is substantially dense that the supplemental note is required rather than the main text standing alone.

Minor comments:

1. In the supplemental note, Table 1 defines the parameters used in the two approaches. Approach 2 defines n, k and p in terms of mutated residues rather than variants. Since the VES calculation ultimately uses variants, it would be worth to include another column of parameter definitions with n, k, and p in terms of variants.

2. The bubble sizes in Figure 5 corresponding to adjusted p-values are difficult to distinguish. In particular, the difference of sizes between 0.05 and 0.25 bubbles is negligible while it is a large difference in terms of significance.

Rev. 2:

The authors have responded in detail to prior questions and critiques, and have improved several aspects of the manuscript. I would still strongly argue for clarifying the manuscript to highlight the primary points in the results section as it is still difficult to follow the logic of the manuscript and understand the most important aspects of the work. In the response, the authors argue that the two main points are "the unbiased comparison of structure and pathway features targeted by missense variants, and the examination of proteomics features in distinguishing missense variants", but I don't feel like either of these primary points are addressed in the six sections of the results. As it reads currently, the authors present and validate their approach with oncogene and tumor suppressor gene analyses in 4.1, then jump to pathway-level analyses in 4.2, then back to protein-feature analyses in 4.3... if the authors wish to highlight the methodological strengths of their approach, tighten the results to specifically highlight all the ways in which the VES score is finding patterns that are supported by prior literature, either with specific protein examples, or with general trends in domains. This could be followed by sections that enumerate unexpected findings that are novel discoveries of the approach, followed with pathway-level enrichments. I only suggest this structure because it would clarify what the authors are presenting as NEW discoveries versus the findings that are largely congruent with what would be expected based on biochemistry/biology. I would also suggest renaming results section headers to more specifically highlight the new knowledge the authors are adding to the literature with their analyses. As it is currently presented, the authors seem to enumerate various different enrichments with little context on if these were expected or novel.

Rev. 3:

No further comments.

---

## [Editor Report · Decision Letter 3]

17 Mar 2021

Dear Franca,

Thank you for submitting your revised Methods and Resources entitled "Pathogenic missense protein variants affect different functional pathways and proteomic features than healthy population variants" for publication in PLOS Biology. I have now discussed it with the editorial team. 

Based on the reviews, we are now satisfied with your responses to the remaining issues raised by the reviewers and will probably accept this manuscript for publication. However, we would like you to clarify some points on the data that are confusing. While all the data underlying the graphs shown in the figures seem to be included, some of the notes made in the figure legends don't match the data shown in the files. Here are a couple of examples:

- In Fig. 3 legend, you mention that the data can be found in Data S2, but in that file you only specify data for Fig. 4, Fig. 5, Fig. 7, Fig. S4, Fig. S5, Fig. S6, Fig. S8, Fig. S9, Fig. S11, Fig. S17 and Fig. S12

- In Fig. 5 legend, you mention the data can be found in Data S6, but in that file, it’s only specified data for Fig. 3, Fig. 7 and Fig. S10.

We would like you to check all the figure legends and data files carefully and do the following:

- Please include in the manuscript a list of all you data files and specify clearly what figures show the data contained in each of them.

- Please make sure that the correct data file is indicated in the corresponding figure legends.

We expect to receive your revised manuscript within two weeks. 

-  a cover letter that should detail your responses to any editorial requests. 

*Published Peer Review History*

*Early Version*

Sincerely,

Ines

--

Ines Alvarez-Garcia, PhD,

Senior Editor,

PLOS Biology

---

## [Editor Report · Decision Letter 4]

26 Mar 2021

Dear Franca,

On behalf of my colleagues and the Academic Editor, Nicole Soranzo, I am delighted to say that we can in principle offer to publish your Methods and Resources entitled "Pathogenic missense protein variants affect different functional pathways and proteomic features than healthy population variants" in PLOS Biology, provided you address any remaining formatting and reporting issues. These will be detailed in an email that will follow this letter and that you will usually receive within 2-3 business days, during which time no action is required from you. Please note that we will not be able to formally accept your manuscript and schedule it for publication until you have made the required changes.

PRESS

Thank you again for supporting Open Access publishing. We look forward to publishing your paper in PLOS Biology. 

Sincerely, 

Ines

--

Ines Alvarez-Garcia, PhD 

Senior Editor 

PLOS Biology